# BAT: Backbone Augmented Training for Adaptations

## Abstract

Adaptations have enabled efficient training for large backbone models such as diffusion models for image generation and transformer-based language models. While various adaptation techniques aim to maximize performance with minimal computational resources, limited data often leads to challenges like overfitting, mode collapse, or hallucinations. Recently, a promising solution has emerged in the form of augmenting adapter datasets using data originally employed to train backbone models. While this approach has shown potential as a breakthrough, it often lacks a solid theoretical foundation or well-defined standards for controllability. To address these limitations, we establish a comprehensive theoretical framework for Backbone Augmented Training (BAT). Furthermore, we provide both theoretical and experimental evidence demonstrating that BAT achieves a faster convergence rate to optimal adaptation parameters compared to conventional adaptation methods. Our results underscore the potential of backbone augmentation to significantly improve performance, especially when coupled with an effective and well-designed data selection schema.

## 1 Introduction

Recently, large foundation models (Brown et al., 2020; Rombach et al., 2022; Meta, 2024; Peebles & Xie, 2023; Sauer et al., 2024) have demonstrated exceptional performance across various tasks. To adapt these models for specific downstream tasks, researchers have introduced a variety of adaptation techniques. These approaches typically involve updating only a small portion of the model parameters—some leveraging rank decomposition (Hu et al., 2021; Dettmers et al., 2023; Liu et al., 2024) of the backbone weights, while others employing fixed text embeddings (Ruiz et al., 2023a; Gal et al., 2022) to maintain identity consistency in image generation.

Despite the success of large models in various downstream tasks, acquiring data for certain tasks remains highly challenging (Lee et al., 2023; Sainz et al., 2023; Gholami & Omar, 2023). The scarcity of data leads to various complications, such as model overfitting (Ruiz et al., 2023b; Pascual et al., 2024; Salman & Liu, 2019), model collapse (Thanh-Tung & Tran, 2020), or hallucination (Luo et al., 2021b). These challenges highlight the critical importance of obtaining sufficient amount of data.

To this end, researchers came up with leveraging the data used to train backbone models. For instance, DreamBooth (Ruiz et al., 2023a) incorporates regularization images generated from the backbone model's distribution. Additionally, datasets commonly used for training diffusion models (Lin et al., 2015; Schuhmann et al., 2022; Bai et al., 2023) and fine-tuned language models (Taori et al., 2023a; Wang et al., 2023; Zhou et al., 2023; Chaudhary, 2023) are often publicly accessible, prompting communities such as jiwenji (2024) and StabilityAI (2024) to heuristically augment adaptation data using backbone data, occasionally yielding positive results.

However, these heuristic methods often lack a clear understanding of how backbone data augmentation enhances model performance. As a result, improving adapter performance using backbone data has largely relied on chance. To address this, in this paper, we first establish the mathematical foundation of Backbone Augmented Training (BAT) and demonstrate the potential of backbone data in adapter training. Beyond theoretical validation, we aim to show through extensive experiments that BAT consistently outperforms non-augmented training under various conditions.

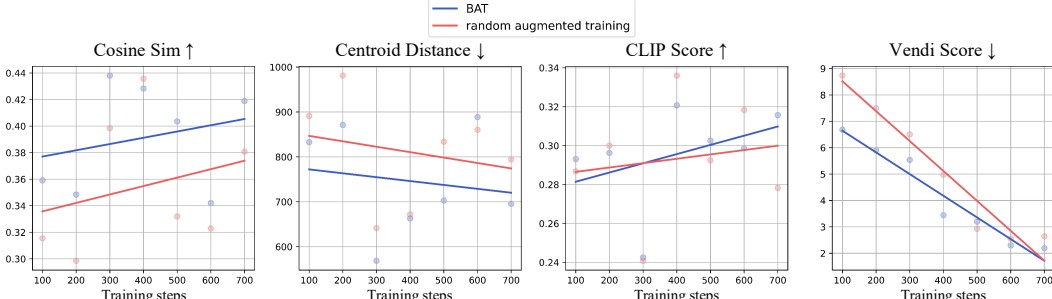

Figure 1: **Personalization Metric Comparison between BAT and Random Augmented Training.** This figure displays the trend of various personalization metrics measured each 100 steps using DreamBooth. As fluctuation of metrics is common in adaptation training, we show the trend line of over all scores. One can observe that all metrics favor BAT in standard personalization metrics.

To support BAT with a solid mathematical foundations, we first adopt reasonable mathematical assumptions proposed in (Kolossov et al., 2023). Based on these assumptions, we formulate two key propositions. The first proposition demonstrates that a BAT-trained adapter converges to an adapater with optimal parameters, justifying the use of backbone data in adaptations. The second proposition offers a fundamental condition that controls the convergence rate of BAT-trained adapters. This proposition highlights the potential of BAT, when combined with effective data selection methods, to surpass accustomed adaptations such as DreamBooth (Ruiz et al., 2023a), LoCon (Yeh et al., 2023), LoRA (Hu et al., 2021), and DoRA (Liu et al., 2024).

Beyond theoretical arguments, we explore the practicality of BAT through experiments across diverse base models, adapters, datasets, and evaluation metrics. Including Fig. 1, the results indicate that with effective data selection, BAT consistently outperforms both random augmentations and standard adaptation methods. Furthermore, our experiments implies that even in scenarios where backbone data is unavailable, performing augmentation using data that follows the backbone model's output distribution still achieves significant performance improvements.

To sum up, the contributions of our paper are as follows:

- We introduce and mathematically define *Backbone Augmented Training for adaptations* and propose Proposition 1 and Proposition 2 to analytically prove that Backbone Augmented Training converges toward the optimal adaptation parameters faster than conventional adaptation training.

- Through experiments, we demonstrate that Backbone Augmented Training consistently outperforms conventional adaptation training across various real-world scenarios. Furthermore, we show that it can still achieve superior performance even in the absence of backbone data or an effective data selection scheme.

## 2 PRELIMINARIES

In this section, we briefly discuss the details of the adaptations used in this study. Also, we define a few notations and concepts behind our experimental approaches.

**Adaptations.** Fine-tuning a large-scale model to solve a downstream task is extremely expensive. To mitigate this challenge, researchers came up with methods that train a small portion of parameters, also known as adaptations. Adaptation methods are widely distinguished as additive fine-tuning (Houlsby et al., 2019; Li & Liang, 2021), selective fine-tuning (Zaken et al., 2021; Guo et al., 2020), reparameterized fune-tuning (Aghajanyan et al., 2020; Karimi Mahabadi et al., 2021). In the following part, we introduce eminent types of adaptations.

**LoRA.** Low-Rank Adaptation (Hu et al., 2021) has gained significant attention among early adaptations for its ability to efficiently train a small portion of parameters through weight decomposition, without any additional inference burden. Specifically, given a pretrained weight matrix $\boldsymbol{W}_0 \in \mathbb{R}^{d \times k}$, LoRA decomposes the weight update $\Delta \boldsymbol{W} \in \mathbb{R}^{d \times k}$ into the product $\boldsymbol{B}\boldsymbol{A}$ to get the adapted matrix

$W = W_0 + \Delta W$. Here, $B \in \mathbb{R}^{d \times r}$ and $A \in \mathbb{R}^{r \times k}$ with $r \ll \min\{d, k\}$. Despite utilizing only a small set of parameters, LoRA achieves performance comparable to full fine-tuning, and in certain benchmarks, even surpasses it. Based on the strong performance of LoRA, several variants emerged including DoRA (Liu et al., 2024; Dettmers et al., 2023) for language models. Others applied this decomposition method in generative models such as diffusion models (Rombach et al., 2022; Song et al., 2022; Ho et al., 2020) like LoCon, LoHA and LoKr (Yeh et al., 2023). However, lack of data can cause overfitting and hallucinations even with this adaptation.

**DreamBooth.** DreamBooth (Ruiz et al., 2023a) is also an adapter for diffusion model which suggests rare-token identifiers to regenerate objects with identical features. Diffusion models before this adaptation had weak capacity in generating same identity repeatedly. For example, generating a famous movie character, a certain cat, over and over again ended up with bunch of cats with different colors and kinds with former methods. Preventing this and achieving the task is called *personalization*. Some attempted to shift the text token in embedding space (Gal et al., 2022), and from this, DreamBooth continues to inject identities in the generation weights with newly defined prior preservation loss. To utilize this loss function, a regularization dataset must be synthesized often much greater in size than the adaptation dataset which can be demanding in practical usage.

**Data Selection.** Recent adaptation users have selected data from the backbone models to mitigate the insufficiency in adaptation data. (jiwenji, 2024; StabilityAI, 2024). However, this method does not show consistent results since they select the backbone data with heuristic and random manner. We name this method as *random augmented training* in this study. However, data selection is an active research topic as it still remains as a crucial part of training models (Zhao et al., 2024; Qin et al., 2024; Wang et al., 2024). The study Kolossov et al. (2023) introduces schemes to select unlabeled data for weakly supervised learning. They use perfect surrogate models that follow the distribution of the full sample whereas imperfect ones do not. The authors develop these schemes from influence functions (Ting & Brochu, 2017; Wang et al., 2021) and leveraging score methods (Ma et al., 2014), and it is notable that the scheme application gives better results than full sample training. Former methods directly applied their score to the loss function to eliminate the impact of unwanted data, but random augmented method simply adds backbone data from their training batch. See Sec. C for further details.

## 3 BACKGROUND

Challenges in adaptation training are often related to acquiring adaptation data. Even though adaptations work well with smaller datasets, the main purpose of adaptation in facilitating a downstream task is often more specific than fine-tuning tasks. Furthermore, some of them aim to personalize the latest identities (Ruiz et al., 2023a; Gal et al., 2022), which make adaptation data extremely rare.

So, we suggest Backbone Augmented Training (BAT), which enhances the adaptation dataset with backbone model training data with theory-based conditions to affirm its benefits.

Within this part, we introduce the notations that will be used consistently throughout the following sections. Then, we demonstrate the mathematical background of adaptation that is newly established. Finally, we show the definitions regarding our method.

### 3.1 BASIC NOTATIONS

For standard notations, we denote the consistency of random variables as $X_n \xrightarrow{P} X$. Using the notation $p$–lim which also implies the consistency of random variables, we define probabilistic asymptotic as:

$$X_n = o_P(a_n) \iff p\text{–}\lim_{n \to \infty} \frac{|X_n|}{a_n} = 0. \tag{1}$$

The notation for almost sure convergence will be noted as:

$$X_n \xrightarrow{a.s.} X \iff \lim_{n \to \infty} P(X_n = X) = 1. \tag{2}$$

Lastly, for some matrices $X$ and $Y$, we denote $X \succeq Y$ if $X - Y$ is positive semi-definite, and $X \succ Y$ if it is positive definite.

Now, for a parameter space $\Theta$ and an estimator $\boldsymbol{\theta}_n \in \Theta$, we define an empirical risk function $R_n : \Theta \to \mathbb{R}$ as:

$$R_n(\boldsymbol{\theta}_n) := \frac{1}{n} \sum_{i=1}^{n} \mathcal{L}_i^{\boldsymbol{\theta}} \iff R_n^{\boldsymbol{\theta}} = R_n(\boldsymbol{\theta}_n), \tag{3}$$

where $\mathcal{L}_i^{\boldsymbol{\theta}} := \mathcal{L}(Y_i, f(X_i; \boldsymbol{\theta}_i))$. $\mathcal{L}$ represents the loss function of the parameters and $i$ reflects the training steps where $f$ is the model. Here, $X$ and $Y$ represent the sampled input and label in model training. We presume the sampling is deterministic as we denote them $\boldsymbol{x}$ and $\boldsymbol{y}$.

After this, by the law of large numbers, we can define some $R$ for $R_n \xrightarrow{P} R$. We set $\hat{\boldsymbol{\theta}}_n$ to be the nearly minimizing estimator that satisfies the following condition:

$$R_n(\hat{\boldsymbol{\theta}}_n) \leq \inf_{\boldsymbol{\theta} \in \Theta} R_n(\boldsymbol{\theta}_n) + o_P(1). \tag{4}$$

Recall that every risk in this study uses sampled sets to optimize their corresponding models. We need to define the total risk to discuss the convergence throughout the whole sample. We can achieve this with a simple expectation to continue this argument:

$$R(\boldsymbol{\theta}) := \mathbb{E}\mathcal{L}(\boldsymbol{y}, f(\boldsymbol{x}; \boldsymbol{\theta})), \tag{5}$$

respect to $(\boldsymbol{x}, \boldsymbol{y}) \sim P(\cdot)$ which makes $\mathcal{D}^{\mathrm{B}}$ and $\mathcal{D}^{\mathrm{A}}$ i.i.d. subsamples from their own distributions. $P(\cdot)$ denotes some given distribution for $(\boldsymbol{x}, \boldsymbol{y})$.

## 3.2 MATHEMATICS ON ADAPTATIONS

Every adaptation method begins with initialization from its backbone model. Using B and A as abbreviations for the backbone and adaptation, we denote the backbone model parameters as $\boldsymbol{\theta}^{\mathrm{B}} \in \Theta^{\mathrm{B}}$ and the combined backbone and adapter parameters as $\boldsymbol{\theta}^{\mathrm{A}} \in \Theta^{\mathrm{A}}$, respectively. Then, loading an initialized adapter over the backbone model can be expressed using a continuous function $g$, that is, $\boldsymbol{\theta}^{\mathrm{A}} := g(\boldsymbol{\theta}^{\mathrm{B}}) \in \Theta^{\mathrm{A}}$. Denoting $\boldsymbol{\theta}^{\mathrm{A}} \backslash \boldsymbol{\theta}^{\mathrm{B}}$ as the parameters exclusive to the adapter, note that $0 < \dim(\boldsymbol{\theta}^{\mathrm{A}} \backslash \boldsymbol{\theta}^{\mathrm{B}}) < \dim(\boldsymbol{\theta}^{\mathrm{B}})$ holds. Typically, while adaptations may introduce more parameters than the backbone model, the backbone model itself is frozen, allowing only a small subset of parameters to be updated. Thus, as the training step $n$ progresses and the $\hat{\boldsymbol{\theta}}_n^{\mathrm{A}}$ are updated toward their optimal values $\boldsymbol{\theta}^{\mathrm{A}^*}$, the parameter update is described as: $(\hat{\boldsymbol{\theta}}^{\mathrm{A}} \backslash \boldsymbol{\theta}^{\mathrm{B}})_{n+1} = (\hat{\boldsymbol{\theta}}^{\mathrm{A}} \backslash \boldsymbol{\theta}^{\mathrm{B}})_n + \Delta(\boldsymbol{\theta}^{\mathrm{A}} \backslash \boldsymbol{\theta}^{\mathrm{B}})_n$.

Let the backbone model be pre-trained with the dataset $\mathcal{D}^{\mathrm{B}}$ via empirical risk minimization. Suppose the dataset $\mathcal{D}^{\mathrm{A}}$ be a training set for the adaptation, usually constructed by the trainer. The size of the datasets is noted as $N := |\mathcal{D}^{\mathrm{B}}|$ and $n := |\mathcal{D}^{\mathrm{A}}|$, respectively, and $n \ll N$ again by adaptations' nature. We denote the model as $f(\cdot; \boldsymbol{\theta}) : \mathbb{R}^{\boldsymbol{p}} \to \mathbb{R}^{\boldsymbol{d}}$ and the loss function as $\mathcal{L} : \mathbb{R}^{\boldsymbol{d}} \times \mathbb{R}^{\boldsymbol{d}} \to \mathbb{R}$. Recall that backbones and adaptations commonly share the loss function. Now, set the backbone risk $R_N^{\mathrm{B}}$ as below, utilizing the regularizer function $\Omega : \Theta \to \mathbb{R}$ and constant $\lambda$ to balance the training:

$$R_N^{\mathrm{B}} := \frac{1}{N} \sum_{\boldsymbol{x}, \boldsymbol{y} \in \mathcal{D}^{\mathrm{B}}} \mathcal{L}(\boldsymbol{y}, f^{\mathrm{B}}(\boldsymbol{x}; \boldsymbol{\theta}^{\mathrm{B}})) + \lambda\Omega(\boldsymbol{\theta}^{\mathrm{B}}), \quad \boldsymbol{\theta}^{\mathrm{B}^*} := \operatorname*{arg\,min}_{\Theta^{\mathrm{B}}} R_N^{\mathrm{B}}. \tag{6}$$

On the other hand, adaptation risk $R_n^{\mathrm{A}}$ is defined as:

$$R_n^{\mathrm{A}} := \frac{1}{n} \sum_{\boldsymbol{x}, \boldsymbol{y} \in \mathcal{D}^{\mathrm{A}}} \mathcal{L}(\boldsymbol{y}, f^{\mathrm{A}}(\boldsymbol{x}; \boldsymbol{\theta}^{\mathrm{A}})) + \lambda\Omega(\boldsymbol{\theta}^{\mathrm{A}}), \quad \boldsymbol{\theta}^{\mathrm{A}^*} := \operatorname*{arg\,min}_{\Theta^{\mathrm{A}}} R_n^{\mathrm{A}}. \tag{7}$$

For the adaptation risk, one should understand that $\mathcal{D}^{\mathrm{B}} \cap \mathcal{D}^{\mathrm{A}} = \emptyset$. This shows that some data in $\mathcal{D}^{\mathrm{B}}$ will make the adaptation risk diverge from the optimal point $\boldsymbol{\theta}^{\mathrm{A}^*}$ while some have the possibility to make the risk converge to it. Consequently, the adaptation risk possesses independent characteristics from the backbone risk, meaning that not all composite functions between two risks always reflect the actual performance of adaptations.

### 3.3 DEFINITIONS

Now, we construct the definitions for Backbone Augmented Training. We start this by introducing a composite empirical risk. Then, the limit value of the proportion of backbone data and adaptation data follows before the asymptotic coefficient of our method.

**Definition 1.** *Backbone augmented training risk on an adaptation is defined as*

$$R_k^{\text{bat}|A} := \frac{1}{k} \sum_{\boldsymbol{x}, \boldsymbol{y} \in \mathcal{D}^{\text{bat}}} \mathcal{L}(\boldsymbol{y}, f^A(\boldsymbol{x}; \boldsymbol{\theta}^{\text{bat}})) + \lambda \Omega(\boldsymbol{\theta}^{\text{bat}}), \tag{8}$$

*for some* $\mathcal{D}^{\text{bat}} = \mathcal{D}^{B'} \cup \mathcal{D}^A$ *where* $\varnothing \neq \mathcal{D}^{B'} \subset \mathcal{D}^B$. *Also,* $k = |\mathcal{D}^{\text{bat}}|$ *and* $\hat{\boldsymbol{\theta}}_1^{\text{bat}} = \hat{\boldsymbol{\theta}}_1^A$.

First, the notation bat|A stands for the application of BAT in the adapter A. $\hat{\boldsymbol{\theta}}_1^{\text{bat}} = \hat{\boldsymbol{\theta}}_1^A$ means that both our method and adaptations are initialized from the same weights. This definition denotes the our method's risk built on the entire adaptation data and some of the backbone data. We will demonstrate in the following section that this risk always increases the performance of adaptations with the application of the next proposition, unlike common composite risks.

**Definition 2.** *Backbone augmentation ratio is denoted as* $n/k \to \gamma \in (0, 1)$.

This ratio essentially shows the proportion of adaptation data and backbone data used in our method. In this definition, we use convergence to derive the ratio and adopt it in our proposition based on asymptotic.

Lastly, following the format of former studies regarding estimators, we continue our augrments by applying asymptotic error coefficients. We first define the coefficients related to the weighted quadratic error $||\hat{\boldsymbol{\theta}} - \boldsymbol{\theta}^*||_{\boldsymbol{S}}^2 := \langle \hat{\boldsymbol{\theta}} - \boldsymbol{\theta}^*, \boldsymbol{S}(\hat{\boldsymbol{\theta}} - \boldsymbol{\theta}^*) \rangle$, where $\boldsymbol{S} \in \mathbb{R}^{\dim(\Theta) \times \dim(\Theta)}$ being $\boldsymbol{I}$ gives a simple Euclidean inner product when $R_N$ is twice differentiable. Additionally, $\boldsymbol{S} = \nabla_{\boldsymbol{\theta}}^2 R_N$ would result the total risk achieved from the iteration of entire epoch of $\mathcal{D}^B$. See Kolossov et al. (2023) for more detailed structure.

Then, we denote an asymptotic error coefficient as $\rho_{\text{B}}(\boldsymbol{S}) := p\text{-}\lim_{N \to \infty} N ||\hat{\boldsymbol{\theta}}^B - \boldsymbol{\theta}^{B^*}||_{\boldsymbol{S}}^2$, with the backbone risk in this case when $\hat{\boldsymbol{\theta}}^B$ refers to a nearly minimizing estimator for $\boldsymbol{\theta}^{B^*}$.

**Definition 3.** *Backbone augmented coefficient on an adaptation is defined as*

$$\rho_{\text{bat}|A}(\boldsymbol{S}) := p\text{-}\lim_{k \to \infty} k ||\hat{\boldsymbol{\theta}}^{\text{bat}} - \boldsymbol{\theta}^{A^*}||_{\boldsymbol{S}}^2. \tag{9}$$

This coefficient may or may not converge depending on the limit of the estimator. If the coefficient's value remains as a real value, we can ensure that the estimator converges to the optimal parameters.

Also, let $\boldsymbol{H}^B(\boldsymbol{x})$ denote the conditional Hessian matrix $\mathbb{E}[\nabla_{\boldsymbol{\theta}}^2 \mathcal{L}^{\boldsymbol{\theta}^{B^*}} | \boldsymbol{x}]$ for parameters of the backbone risk. This matrix is useful in representing the parameter update in optimization with respect to related variables. If the notation $B$ is replaced, then the matrix is associated with another model and its empirical risk.

## 4 BACKBONE AUGMENTED TRAINING FOR ADAPTATIONS

### 4.1 ASSUMPTIONS

Herein, we propose the four assumptions about the nature of the backbone and adaptation risks that are basic in asymptotic estimation theories (Kolossov et al., 2023). The fifth one is our novel assumption as we introduce our method's risk in this study for the first time.

**Assumption 1.** $R^B$ *and* $R^A$ *are minimized uniquely at* $\boldsymbol{\theta}^{B^*}$ *and* $\boldsymbol{\theta}^{A^*}$ *respectively.*

**Assumption 2.** $\mathcal{L}^B$ *and* $\mathcal{L}^A$ *are both greater than zero and lower semi-continuous always. Moreover, for every* $\boldsymbol{u} \in \mathbb{S}^{\dim(\Theta^B)-1}$ *and* $g(\boldsymbol{u}) \in \mathbb{S}^{\dim(\Theta^A)-1}$, *define* $\mathcal{L}_\infty^B$ *and* $\mathcal{L}_\infty^A$ *both in* $\overline{\mathbb{R}}_{\geq 0}$ *as:*

$$\mathcal{L}_\infty^B(\boldsymbol{u}; \boldsymbol{x}, \boldsymbol{y}) := \liminf_{\substack{||\boldsymbol{\theta}|| \to \infty \\ \boldsymbol{\theta}/||\boldsymbol{\theta}|| \to \boldsymbol{u}}} \mathcal{L}^B, \quad \mathcal{L}_\infty^A(g(\boldsymbol{u}); \boldsymbol{x}, \boldsymbol{y}) := \liminf_{\substack{||\boldsymbol{\theta}|| \to \infty \\ \boldsymbol{\theta}/||\boldsymbol{\theta}|| \to g(\boldsymbol{u})}} \mathcal{L}^A, \tag{10}$$

*and suppose* $\inf_{\boldsymbol{u}} \mathbb{E}\mathcal{L}^{\mathrm{B}}_{\infty} > R(\boldsymbol{\theta}^{\mathrm{B}^*})$ *and* $\inf_{g(\boldsymbol{u})} \mathbb{E}\mathcal{L}^{\mathrm{A}}_{\infty} > R(\boldsymbol{\theta}^{\mathrm{A}^*})$.

*Assumption 3. Both* $\mathcal{L}^{\boldsymbol{\theta}^{\mathrm{B}}}$ *and* $\mathcal{L}^{\boldsymbol{\theta}^{\mathrm{A}}}$ *are differentiable at* $\boldsymbol{\theta}^{\mathrm{B}^*}$ *and* $\boldsymbol{\theta}^{\mathrm{A}^*}$ *respectively for* $\mathbb{P}$-*almost all* $(\boldsymbol{y}, \boldsymbol{x})$. *Further, for a neighborhood* $U$ *of* $\boldsymbol{\theta}^{\mathrm{B}^*}$ *or* $\boldsymbol{\theta}^{\mathrm{A}^*}$, *as*

$$\mathbb{E} \sup_{\boldsymbol{\theta}_1 \neq \boldsymbol{\theta}_2 \in U} \left[ \frac{|\mathcal{L}(\boldsymbol{\theta}_1) - \mathcal{L}(\boldsymbol{\theta}_2)|}{||\boldsymbol{\theta}_1 - \boldsymbol{\theta}_2||_2^2} \right] < \infty. \tag{11}$$

*Assumption 4.* $R^{\mathrm{B}}$ *and* $R^{\mathrm{A}} \in C^2$ *with existing* $\boldsymbol{H}^{\mathrm{B}}(\boldsymbol{x}), \boldsymbol{H}^{\mathrm{A}}(\boldsymbol{x}) \succeq \boldsymbol{0}$.

*Assumption 5. For any neighborhood* $U^n$ *of* $\boldsymbol{\theta}^{\mathrm{A}^*}$ *where* $\hat{\theta}^{\mathrm{bat}}_n \in U^n$, *any* $R^{\mathrm{A}}(\boldsymbol{\theta}) - R^{\mathrm{bat}}(\boldsymbol{\theta}) \neq R^{\mathrm{A}}(\boldsymbol{\theta}^{\mathrm{A}^*}) - R^{\mathrm{bat}}(\boldsymbol{\theta}^{\mathrm{A}^*})$ *for any* $\boldsymbol{\theta} \in \Theta^{\mathrm{A}}$ *except* $\boldsymbol{\theta} = \boldsymbol{\theta}^{\mathrm{A}^*}$.

Assumption 1 states that the risks have unique minimum values which is a common setting in theoretical proofs (Kolossov et al., 2023; Ai et al., 2021). Assumption 2 means that the risks are continuous and their value is finite. The third and forth ones assume both backbone and adaptation risks are differentiable and convex. These assumptions are weak conditions that are satisfied when we assume that the model is learnable. Finally, the fifth assumption presumes that the our method's risk is a smooth function when we map it near the domain that includes the adaptation's optimal parameter.

## 4.2 MAIN PROPOSITIONS

Upon the assumptions in Sec. 4.1, we present two propositions regarding our method's risk. Due to the page limit, we leave the proofs in Sec. A.4 and Sec. A.5.

**Proposition 1** *(Validity of Backbone Augmented Training).*
*Suppose the assumptions in Sec. 4.1 hold. Then, for any* $\boldsymbol{S} \in \mathbb{R}^{dim(\Theta^A) \times dim(\Theta^A)}$ *that is symmetric,* $\rho_{bat|A}(\boldsymbol{S})$ *exists.*

Proposition 1 is mainly about the backbone augmentation coefficient. This shows the rate of convergence to the optimal adaptation. The existence of this coefficient $\rho_{bat|A}$ implies that the our adaptation represented by the coefficient will eventually converge to its optimal parameters. Thus, the proposition is named the validity of BAT. By utilizing this proposition, we justify BAT specifically in DreamBooth (Ruiz et al., 2023a) and LoRA (Hu et al., 2021) in Sec. A.6.

**Proposition 2** *(Condition for Backbone Augmented Training).*
*Let* $\mathcal{D}^{bat} \cap \mathcal{D}^{\mathrm{B}} = \mathcal{D}^{\mathrm{B}'}$, *and* $\boldsymbol{H}^{bat} = \mathbb{E}[\nabla^2_{\boldsymbol{\theta}} \mathcal{L}^{bat|A} | \boldsymbol{x}] \iff (\boldsymbol{x}, \boldsymbol{y}) \in \mathcal{D}^{\mathrm{B}'}$. *If*

$$\gamma ||(\boldsymbol{H}^{bat|A})^{-1} \sum_{\mathcal{D}^{bat}} \nabla_{\boldsymbol{\theta}} \mathcal{L}^{bat|A}|| \leq ||(\boldsymbol{H}^{bat|A} - \boldsymbol{H}^{bat})^{-1} \sum_{\mathcal{D}^{\mathrm{A}}} \nabla_{\boldsymbol{\theta}} \mathcal{L}^{bat|A}|| + o_P(1) \tag{12}$$

*holds with respect to any* $\boldsymbol{\theta} \in (\boldsymbol{\theta}^{\mathrm{A}} \cap \boldsymbol{\theta}^{\mathrm{B}})$, *then* $\rho_{bat|A} \leq \rho_A$ *holds with assumptions of Proposition 1 and unless* $\gamma \to 1$, *the inequality is strict.*

In Proposition 2, we show the basic condition for backbone data that surpass the regular adaptation training. The value on the left side of the inequality is derived from $\mathcal{D}^{bat}$. This proposition is particularly showing that if this value is smaller than the value on the right side, our method will surpass the regular adaptation training. This comparison becomes the key to the data selection of $\mathcal{D}^{bat}$. The mathematical model in Fig. 2 depicts that both risks are separated and BAT parameters are moving in different path in parametric space. Also, the proposition indicates that the brute calculation for data selection requires much lesser computation than the calculation for backbone training as the number of parameters for Hessian matrix shrinks. Furthermore, note that in the proposition, $\boldsymbol{H}^{\mathrm{A}}$ disappeared along the proof. This means that Hessian calculation for the original adaptation is no longer required and tracking $\boldsymbol{H}^{bat|A}$ will be sufficient. This is useful information as Hessian calculation demands heavy computations.

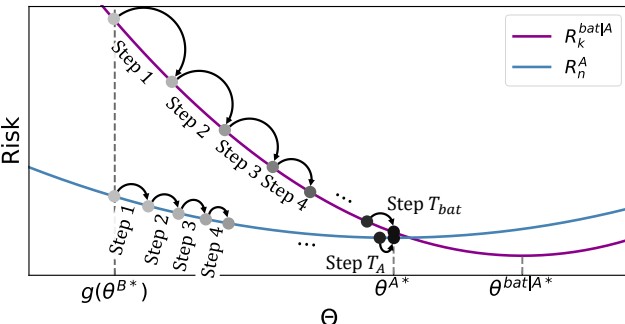

Figure 2: **Visualization of Empirical Risk according to Training Steps.** By Proposition 2, BAT, with a smaller asymptotic error coefficient, reduces risk faster than regular adaptation as training progresses. Therefore, using a risk function with additional backbone data serves as a shortcut to optimize adaptation.

### 4.3 TRAINING AN ADAPTER REGARDING THE PROPOSITIONS

According to Proposition 2, if we successfully select data from the backbone dataset that satisfies the proposition, a BAT-trained adapter will outperform non-augmented adapters. However, as the primary focus of this paper is to demonstrate the potential of the backbone dataset, we conduct our experiments under the assumption that data selection is performed effectively.

First, we train an adapter on $\mathcal{D}^A$ with sufficient amount of training steps and assume the final parameters obtained be the optimal parameters $\theta^{A^*}$. Next, to train the adapter using the BAT approach, we identify data samples from the backbone dataset that satisfy Proposition 2 at each training step. These selected samples are added in the adapter's data batch, and training proceeds accordingly. The detailed training algorithm is elaborated in Sec. C. Since our study focuses on demonstrating the potential of leveraging the backbone dataset for adapter training, the assumption of obtaining optimal parameters precedes the experiments. Developing an advanced data selection algorithm that does not rely on prior knowledge of the optimal parameters remains as our future work.

## 5 EXPERIMENTS

To validate our propositions, we demonstrate that models trained with Backbone Augmented Training (BAT) outperform their counterparts. Specifically, we compare the performance of the BAT-trained model with two alternatives: a model trained exclusively on the $\mathcal{D}^A$ dataset only, and a model trained $\mathcal{D}^{bat}$ but with randomly sampled backbone data, that is, the random augmented training. First we present results of weight difference, a metric suitable for verifying our propositions (Sec. 5.1). Subsequently, we provide benchmark results to show that BAT is also practically applicable in real-world scenarios (Sec. 5.2).

Our goal is to demonstrate that BAT can be effectively applied across various tasks and adaptation methods. To this end, we evaluate its performance on personalization tasks using DreamBooth (Ruiz et al., 2023a) and LoCon from LyCORIS (Yeh et al., 2023), and present results for commonsense reasoning tasks with LLaMA 2-7B (Touvron et al., 2023). Since most language models do not disclose their pre-training datasets (i.e., $\mathcal{D}^B$), we adopted the publicly available model that had undergone further fine-tuning as the backbone model. Further details on training features are mentioned in Sec. B.

### 5.1 VALIDATING BAT WITH WEIGHT DIFFERENCE

Since the satisfaction of Proposition 2 requires Proposition 1 to be satisfied, we focus on validating Proposition 2, which is $\rho_{\text{bat}|A} \leq \rho_A$. Note that in Proposition 2, the notation in equation 9 regarding $\rho_{\text{bat}|A}$ is converted to a Hessian expression as both of them involve measuring the difference between the parameters of BAT-trained model and those of the optimal model. We refer to this metric $||\boldsymbol{H}^{-1} \sum_{\mathcal{D}} \nabla_{\boldsymbol{\theta}} \mathcal{L}||$ as the weight difference, and show that it decreases progressively as the training steps increase.

### 5.1.1 BAT VERSUS RANDOM AUGMENTATION

We show that BAT with Proposition 2 is better than random augmented training. First, we divide $\mathcal{D}^A$ into two portions, with one portion being larger than the other. Then, we train a DreamBooth adapter with a the larger portion with a sufficient amount of training iteration, and assume the resulting model parameters as the optimal parameters $\theta^*$. Subsequently, we start training two other adaptations, one using BAT and one with random augmentation. During training, we measured the weight different to assess how close the model parameters $\theta$ were to the optimal parameters $\theta^*$. Note that the small and large datasets do not share any data samples.

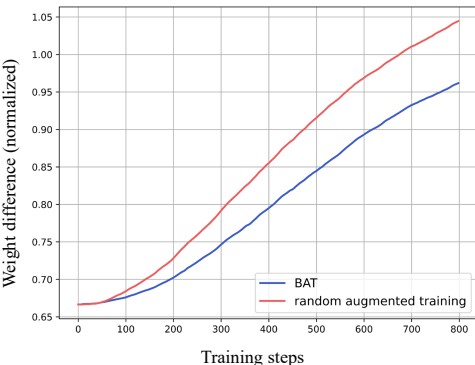

Figure 3: **Full Step Comparison of Weight Difference between BAT and Random Augmented Training.** The graph shows that when BAT meets the condition of Proposition 2, the weight difference is smaller than random augmented training throughout the entire training. We intentionally use limited size of adaptation datasets to reproduce the lack of data that is common among the end users.

**Results.** As shown in Fig. 3, we repeatedly observe many cases that the random augmented training results in a slower convergence rate than our scheme until the same optimal iteration steps. This supports our propositions, implying that along with the optimal steps our scheme surpasses the random selection method in convergence to optimal parameters.

### 5.1.2 BAT VERSUS NON-AUGMENTED TRAININGS

In this experiment, we assert that BAT outperforms non-augmented adapter training. Recall that, as mentioned earlier, it has been discovered that expanding datasets demonstrate a certain level of effectiveness. Therefore, for this experiment, we impose a more challenging setup. We first train an adapter on $\mathcal{D}^A$, assuming that the resulting model possesses the optimal parameters $\theta^{A^*}$. Then, we train another adapter with a same initial parameters, but applying backbone augmentation on $\mathcal{D}^A$. We again measure how far the parameters of the adapter from $\theta^{A^*}$, at each step $n$. This setting is more challenging than the experiment in Sec. 5.1.1, as $\theta^A \to \theta^{A^*}$ is guaranteed while $\hat{\theta}^{bat} \to \theta^{A^*}$ is not.

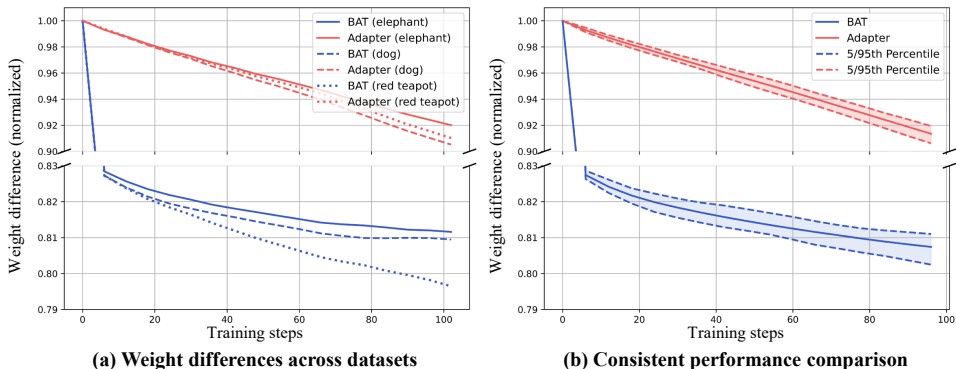

(a) Weight differences across datasets   (b) Consistent performance comparison

Figure 4: **Initial Step Comparisons Between BAT with DreamBooth.** Blue and red represent the convergence rates of BAT and the regular adapter, respectively. (a) and (b) depict results across different datasets and random seeds.

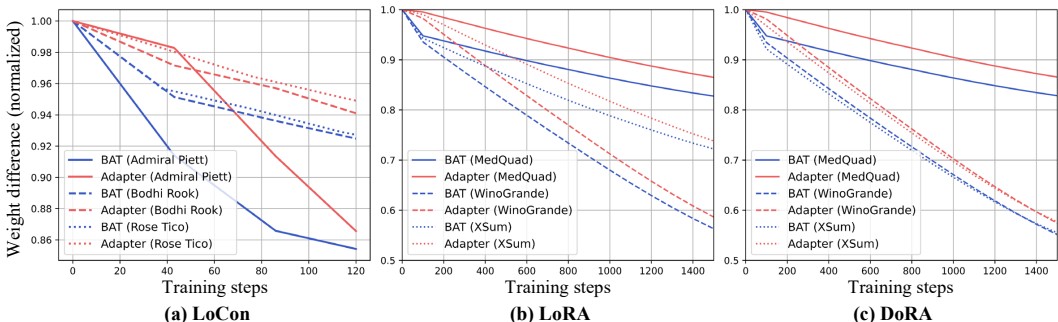

(a) LoCon  (b) LoRA  (c) DoRA

Figure 5: **Initial Step Comparison with Other Adaptations.** This figure shows the results of the experiment from Sec. 5.1.2 using LoCon (Yeh et al., 2023), LoRA (Hu et al., 2021), and DoRA (Liu et al., 2024), exhibiting a similar pattern to Fig. 4. The weight differences were calculated with in certain interval steps across the 200 and 1400 total steps correspondingly.

**Results.** Fig. 4 illustrates that BAT achieves a higher convergence rate compared to DreamBooth across different datasets and various seeds, respectively. Moreover, Fig. 5 indicates that BAT outperforms other various adaptations without incorporating any backbone data. These results suggest that, despite the rigor of the setting, our concept surpassed regular training under varying conditions at certain steps. However, in the final stage of training, our scheme fails to find backbone data that meets the condition of Proposition 2. This is because, in our setting, $\theta^A$ is guaranteed to converge to $\hat{\theta}^{A^*}$, making it increasingly difficult for $\hat{\theta}^{\text{bat}}$ to approach $\theta^{A^*}$ more closely than $\theta^A$ after a certain point.

## 5.2 EVALUATING BAT WITH BENCHMARKS

### 5.2.1 BENCHMARK TEST

In Sec. 5.1, we validated our propositions with carefully designed settings suitable for the validation. Now, we demonstrate that our method improves the capacity of adaptations in more practical scenarios. To show that BAT achieves a faster convergence rate compared to regular adaptations, we evaluate benchmark scores for BAT and standard adaptations at earlier training steps. Specifically, we evaluate 8 benchmark (Clark et al., 2019; Bisk et al., 2019; Lu et al., 2022; Zellers et al., 2019; Sakaguchi et al., 2021; Clark et al., 2018; Luo et al., 2021a) scores for LLaMA 2-7B with LoRA adaptations at the first epoch. Additionally, standard metric scores for diffusion adaptations are assessed at 300 to 700 steps.

| | Cosine Sim ↑ | Centroid Distance ↓ | CLIP ↑ | Vendi ↓ |
|---|---|---|---|---|
| DreamBooth (Ruiz et al., 2023a) | 0.386 | 797.78 | 0.267 | 4.812 |
| + BAT | **0.418** | **695.67** | **0.315** | **2.191** |
| LoCon (Yeh et al., 2023) | 0.5427 | 82.35 | 0.4884 | 1.8471 |
| + BAT | **0.5502** | **82.48** | **0.4952** | **1.8391** |

| | BoolQ | PIQA | SIQA | HellaSwag | WinoGrande | ARC-c | ARC-e | OBQA |
|---|---|---|---|---|---|---|---|---|
| LoRA | 62.17 | 76.28 | 74.51 | 24.61 | 48.86 | 48.70 | **74.07** | **32.70** |
| + BAT | **65.17** | **80.25** | **77.02** | **73.01** | **51.38** | **53.20** | 71.93 | 42.83 |
| DoRA | 62.17 | 76.50 | 72.36 | 24.41 | 50.28 | 37.54 | **74.96** | **60.80** |
| + BAT | **63.96** | **78.84** | **74.36** | **90.77** | **73.88** | **42.66** | 71.89 | 57.00 |

Table 1: **Comparison of Benchmarks between BAT and Various Adaptations.** For more detailed explanation regarding metrics refer to Sec. D.

**Result.** Fig. 1 shows that our method beats random augmented training throughout the whole training steps in DreamBooth adaptation. Also, Tab. 5.2.1 demonstrates that our method surpasses regular adaptation scores in most of the language model benchmarks and image generation measurements. Particularly, the benchmarks, Hellaswag and WinoGrande (Zellers et al., 2019; Sakaguchi et al., 2021), are more responsive to the adaptation's rank decomposition, but BAT mitigates this effect and achieve far better results. On the other hand, for ARC-e and OBQA (Clark et al., 2018; Luo et al., 2021a), as these benchmarks require more task specific knowledge, BAT decreases the downstream performance slightly. These results coincide with the results of Sec. 5.1.2 as the final stage of the former experiment and these benchmarks impose the model to be trained with a more uniform data.

### 5.2.2 INACCESSIBLE BACKBONE DATA

Many large models do not release their training data currently (Brown et al., 2020; Sauer et al., 2024). However, we can always explore their input and output features. With the feature information, we may select open-source data that has similar distributional features in both the data point and dataset perspective. This study does not propose theoretically modified propositions regarding this case, but we investigate this matter by applying similar datasets that are not a part of the backbone dataset. We have executed this experiment with DreamBooth by attaining similar data used in the successful case of BAT training, online.

**Results.** The result shows that similar data still retains our method's effect even when they are not in the backbone data. Our method has selected data from online that satisfies Proposition 2. The result in Tab. 5.2.2 shows better scores than regular adaptation in most cases, but not as favorable as original BAT.

| | Cosine Sim ↑ | Centroid Distance ↓ | CLIP ↑ | Vendi ↓ |
|---|---|---|---|---|
| DreamBooth (Ruiz et al., 2023a) | **0.386** | 797.78 | 0.267 | 4.812 |
| + BAT | 0.365 | **795.78** | **0.291** | **4.722** |

Table 2: **Comparison of Personalization Scores with DreamBooth Using Data Out of Backbone.** This figure depicts using similar data that is not in the backbone dataset may have similar effect with BAT. However, the result is not as consistent as BAT.

## 6 CONCLUSION

Our study introduces and defines Backbone Augmented Training (BAT) in most rigorous way possible. We also conduct experiments to prove our propositions and demonstrate the real world outcomes which shows their alignment and promising results.

**Limitations.** However, the readers must understand that our study is less focused on achieving better performance in adaptations, but suggesting that this idea is very much worthy to investigate for the development of adaptations. In mathematical terms, the convexity and continuity assumptions in the propositions may not be applied to some adaptation architectures. Also, our experimental setting adopts random data sampling before conditional selection which is proven to be inferior to proper selection methods such like Kolossov et al. (2023).

**Future Work.** Many future works are present as our study comprehends broad domains and techniques. First, we propose mathematical improvements on Proposition 2. Like many other optimization problems (Hinton & Salakhutdinov, 2006; Song et al., 2020; Kingma & Welling, 2022), we speculate that the condition to choose helpful backbone data can be more implicit and swift. Also, the development in entire data selection scheme would make the idea more practical and influential. Finally, analysis of the favorable and unsuitable backbone data will provide a more profound understanding of the relationship between adaptations and backbone models.

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

## A    MATHEMATICAL SUPPLEMENTS

### A.1    THEOREM 1

*Assume that the map $\mathcal{L}^{\boldsymbol{\theta}}(\boldsymbol{x}) : \Theta \to \mathbb{R}$ is lower semi-continuous for almost all $\boldsymbol{x}$ which is any input data of the estimator. Then, for any $\boldsymbol{\theta} \in \Theta$,*

$$\mathcal{L}^{\boldsymbol{\theta}}(\boldsymbol{x}) \leq \liminf_{\boldsymbol{\theta}_n \to \boldsymbol{\theta}} \mathcal{L}^{\boldsymbol{\theta}_n}(\boldsymbol{x}), \quad \textit{almost surely.} \tag{13}$$

**Proof of Theorem 1.** We begin by recalling the definition of lower semi-continuity. A function $f : \Theta \to \mathbb{R}$ is lower semi-continuous at $\theta$ if:

$$\liminf_{\theta_n \to \theta} f(\theta_n) \geq f(\theta).$$

This property ensures that the function does not suddenly drop in value near $\theta$. Formally, for any sequence $\theta_n \to \theta$, we have:

$$\liminf_{n \to \infty} f(\theta_n) \geq f(\theta)$$

Given that $\mathcal{L}^{\theta}(x)$ is lower semi-continuous for almost all $x$, we can apply the definition of lower semi-continuity. Specifically, for any $\theta \in \Theta$ and any sequence $\theta_n \to \theta$, it follows that:

$$\mathcal{L}^{\theta}(x) \leq \liminf_{\theta_n \to \theta} \mathcal{L}^{\theta_n}(x).$$

This inequality holds because $\mathcal{L}^{\theta}(x)$ is assumed to be lower semi-continuous.

The term *almost surely* in this context means that the inequality holds for almost all values of $x$ (in a probabilistic or measure-theoretic sense). In other words, there may be a set of measure zero where the inequality does not hold, but this set is negligible.

Thus, for almost every $x$ (except on a set of measure zero), the following inequality holds:

$$\mathcal{L}^{\theta}(x) \leq \liminf_{\theta_n \to \theta} \mathcal{L}^{\theta_n}(x). \qquad \text{almost surely}$$

By combining these observations, we conclude that since $\mathcal{L}^{\theta}(x)$ is lower semi-continuous for almost all $x$, for any sequence $\theta_n \to \theta$, the theorem is proven. $\qquad \square$

### A.2    THEOREM 2

*For any sufficiently small neighborhood $U \subset \Theta$ around $\boldsymbol{\theta}$, if the map $\inf_{\boldsymbol{\theta} \in U} \mathcal{L}^{\boldsymbol{\theta}}(\boldsymbol{x}) : \mathbb{R}^{\boldsymbol{p}} \to \mathbb{R}$ satisfies the condition of Theorem 1, then the map is measurable and $R(\boldsymbol{\theta}) > -\infty$ for $\boldsymbol{\theta}$ that satisfies $\inf_{\boldsymbol{\theta} \in U} \mathcal{L}^{\boldsymbol{\theta}}$.*

**Proof of Theorem 2.** Using Theorem 1 (Sec. A.1), we know that if $\mathcal{L}^{\theta}(x)$ is lower semi-continuous, then for any $\theta \in \Theta$:

$$\mathcal{L}^{\theta}(x) \leq \liminf_{\theta_n \to \theta} \mathcal{L}^{\theta_n}(x) \quad \text{almost surely.}$$

This property guarantees that the function does not suddenly drop in value and behaves well under limits of sequences.

Now, let us analyze the map $\inf_{\theta \in U} \mathcal{L}^{\theta}(x)$, which is the infimum of $\mathcal{L}^{\theta}(x)$ over a neighborhood $U \subset \Theta$ around $\theta$. The function $\mathcal{L}^{\theta}(x)$ is assumed to satisfy the lower semi-continuity condition of Theorem 1 (Sec. A.1).

We now show that the map $\inf_{\theta \in U} \mathcal{L}^{\theta}(x)$ is measurable. Since lower semi-continuous functions are measurable in standard measure theory, we conclude that $\mathcal{L}^{\theta}(x)$ is measurable. Further, the infimum of a collection of lower semi-continuous functions over a compact set is itself lower semi-continuous, and hence measurable.

Next, define $R(\theta) = \inf_{\theta \in U} \mathcal{L}^{\theta}(x)$. We need to show that $R(\theta) > -\infty$. Since $\mathcal{L}^{\theta}(x) \in \mathbb{R}$ is bounded from below and lower semi-continuous on a compact set, the infimum will also be bounded from below. Hence, $R(\theta) > -\infty$.

Thus, the theorem is proven. $\qquad \square$

### A.3 THEOREM 3

*Let the map $\mathcal{L}^{\boldsymbol{\theta}}(\boldsymbol{x}) : \Theta \to \mathbb{R}$ satisfies the conditions for Theorem 1 (Sec. A.1) and 2 (Sec. A.2). Then, for any nearly minimizing estimator $\hat{\boldsymbol{\theta}}_n$ and some globally minimizing parameter $\boldsymbol{\theta}^* \in \Theta^*$ for some global minimum space in case there are multiple or continuous set of globally minimizing parameters, for any $\varepsilon > 0$ and compact set $A \subset \Theta$,*

$$P(\text{dist}(\hat{\boldsymbol{\theta}}_n, \Theta^*) \geq \varepsilon \wedge \hat{\boldsymbol{\theta}}_n \in A) \to 0. \tag{14}$$

**Proof of Theorem 3.**

*Case 1.* For all $\boldsymbol{\theta} \in \Theta$, assume $R(\boldsymbol{\theta}) = \infty$, then by the assumption of nearly minimum and derivation with the law of large number like above, $R_n(\hat{\boldsymbol{\theta}}_n) \leq R(\boldsymbol{\theta}^*) + o_P(1)$. This makes all $R_n(\hat{\boldsymbol{\theta}}_n)$ converge to $\infty$ in probability, letting $\Theta = \Theta^*$ and $\text{dist}(\hat{\boldsymbol{\theta}}_n, \Theta^*) \xrightarrow{P} 0$. Now, for the case where for some $\boldsymbol{\theta}^*$ such that $R(\boldsymbol{\theta}^*) < \infty$, let $U_m \downarrow \boldsymbol{\theta}$ be a diminishing sequence of open neighborhoods around a chosen $\boldsymbol{\theta}$ as their diameters converge to zero. Then, by the assumption of Theorem 2 (Sec. A.2), $R(\boldsymbol{\theta}^*) > -\infty$ when $\mathcal{L}^{\boldsymbol{\theta}^*} = |\mathcal{L}^{\boldsymbol{\theta}^*}|$ for all $X$ and $Y$.

Denote $\mathcal{L}^U(\boldsymbol{x})$ for $\inf_{\boldsymbol{\theta} \in U} \mathcal{L}^{\boldsymbol{\theta}}(\boldsymbol{x})$. The sequence $\mathcal{L}^{U_m}$ is increasing and lower than $\mathcal{L}^{\boldsymbol{\theta}}$ by its definition. Then, by Theorem 1 (Sec. A.1), regarding $\boldsymbol{\theta}_n \to \boldsymbol{\theta}$, as some $\boldsymbol{\theta}' \in U_m \to \boldsymbol{\theta}$, $\mathcal{L}^{U_m}$ is the left-hand limit of $\mathcal{L}^{\boldsymbol{\theta}}$ almost surely. Recall the monotone convergence theorem (Tao, 2011), then by the definition of $R$ which involves expectation and integral, $R^U(\boldsymbol{\theta}_m)$ where $\boldsymbol{\theta}_i$ satisfies $\mathcal{L}^{U_i}$ is also the left-hand limit of $R(\boldsymbol{\theta})$.

*Case 2.* For $\boldsymbol{\theta} \notin \Theta^*$, $R(\boldsymbol{\theta}) > R(\boldsymbol{\theta}^*)$ by definitions. Then, from the proceeded arguments, there exists an open neighborhood $U^{\boldsymbol{\theta}}$ of $\boldsymbol{\theta}$ where $R(\boldsymbol{\theta}) > R(\boldsymbol{\theta}^*)$. This implies that the set $B = \{\boldsymbol{\theta} \in A : dist(\boldsymbol{\theta}, \Theta^*) \geq \varepsilon\}$ is compact as it is covered by the subset of $\{U^{\boldsymbol{\theta}} : \boldsymbol{\theta} \in B\}$.

Let $U^{\boldsymbol{\theta}_1}, U^{\boldsymbol{\theta}_2}, \ldots, U^{\boldsymbol{\theta}_p}$ be such subcovers. By the law of large numbers and definition of $U$,

$$\inf_{j=1,\ldots,p} R_n^U(\boldsymbol{\theta}_j) \leq \inf_{\boldsymbol{\theta} \in B} R_n(\boldsymbol{\theta}) \xrightarrow{a.s.} R(\boldsymbol{\theta}^*) < \inf_j R^U(\boldsymbol{\theta}_j). \tag{15}$$

If $\hat{\boldsymbol{\theta}}_n \in B$, then $\inf_{\boldsymbol{\theta} \in B} R_n(\boldsymbol{\theta})$ is less than or equal to $R_n(\hat{\boldsymbol{\theta}})$ by $B$'s definition. Then by the definition of $\hat{\boldsymbol{\theta}}_n$, $\inf_{\boldsymbol{\theta} \in B} R_n(\boldsymbol{\theta})$ is also less than or equal to $R_n(\boldsymbol{\theta}^*)$ and also less than or equal to $R(\boldsymbol{\theta}^*)$ as $n \to \infty$ by the consistency of $R_n$ covered under the definition of it. So,

$$\{\hat{\boldsymbol{\theta}} \mid \hat{\boldsymbol{\theta}} \in B\} \subset \{\inf_{\boldsymbol{\theta} \in B} R_n(\boldsymbol{\theta}) \leq R(\boldsymbol{\theta}^*) + o_P(1)\}. \tag{16}$$

This means that the probability of the event on the right side, which is the equivalent to the last line of the theorem, converges to zero, proving this theorem. □

### A.4 PROOF OF PROPOSITION 1.

For $||\boldsymbol{\zeta}|| < 1$, define $\boldsymbol{\varphi}(\boldsymbol{\zeta}) = r(||\boldsymbol{\zeta}||)\boldsymbol{\zeta}$ with $r(c) = 1/(1 - c^2)$ to deal with more concentrated parameters than unit parameters, then define the loss for batched adaptation,

$$\mathcal{L}^{\text{bat}|\text{A}}(\boldsymbol{\zeta}; \boldsymbol{x}, \boldsymbol{y}) := \begin{cases} \mathcal{L}^{\text{bat}|\text{A}}(\boldsymbol{\varphi}(\boldsymbol{\zeta}); \boldsymbol{x}, \boldsymbol{y}) \text{ if } ||\boldsymbol{\zeta}|| < 1, \\ \mathcal{L}^{\text{bat}|\text{A}}_{\infty}(\boldsymbol{\zeta}; \boldsymbol{x}, \boldsymbol{y}) \quad \text{if } ||\boldsymbol{\zeta}|| = 1, \end{cases} \tag{17}$$

so that

$$R^{\text{bat}|\text{A}}(\boldsymbol{\zeta}) = \mathbb{E}\mathcal{L}^{\text{bat}|\text{A}}(\boldsymbol{\zeta}, \boldsymbol{x}, \boldsymbol{y}), \quad R_k^{\text{bat}|\text{A}} = k^{-1}\sum_i^k \mathcal{L}^{\text{bat}|\text{A}}(\boldsymbol{\zeta}, \boldsymbol{x}_i, \boldsymbol{y}_i), \tag{18}$$

for $\boldsymbol{\zeta} \in \mathbb{B}^{dim(\Theta^{\text{A}})}(1)$ which is a unit ball in $\Theta^{\text{A}}$ and $(\boldsymbol{x}, \boldsymbol{y}) \in G$. Suppose that

$$\hat{\boldsymbol{\zeta}} := \underset{\boldsymbol{\zeta} \in \mathbb{B}^{dim(\Theta^{\text{A}})}}{\arg\min} (|R_n^{\text{A}}(\boldsymbol{\theta}^{\text{A}^*}) - R_k^{\text{bat}|\text{A}}(\hat{\boldsymbol{\theta}}_k^{\text{bat}})| - |R_n^{\text{A}}(\boldsymbol{\theta}^{\text{A}^*}) - R_k^{\text{bat}|\text{A}}(\boldsymbol{\theta}^{\text{A}^*})|), \tag{19}$$

$$\boldsymbol{\zeta}^* := \underset{\boldsymbol{\zeta} \in \mathbb{B}^{dim(\Theta^{\text{A}})}}{\arg\min} (|R^{\text{A}}(\boldsymbol{\theta}^{\text{A}^*}) - R^{\text{bat}|\text{A}}(\hat{\boldsymbol{\theta}}^{\text{bat}})| - |R^{\text{A}}(\boldsymbol{\theta}^{\text{A}^*}) - R^{\text{bat}|\text{A}}(\boldsymbol{\theta}^{\text{A}^*})|). \tag{20}$$

The second term is unique from Assumption 5. Recall that $\mathcal{L}^{\text{bat}|\text{A}}$ is defined on both $\mathcal{D}^{\text{B}}$ and $\mathcal{D}^{\text{A}}$. We know that $\mathcal{L}^{\text{bat}|\text{A}}$ defined on $\mathcal{D}^{\text{A}}$ is simply $\mathcal{L}^{\text{A}}$ as $\boldsymbol{\theta}^{\text{bat}} \in \Theta^{\text{A}}$ by definition. Thus, the continuity

feature is demonstrated. However, for $\mathcal{L}^{\text{bat}|\text{A}}$ defined on $\mathcal{D}^{\text{B}}$, one has to use the nature of adaptation to depict the lower semi-continuity.

Since $\mathcal{L}$ is a compositional function of $f$, $\boldsymbol{\theta}$, and $(\boldsymbol{x}, \boldsymbol{y})$, showing $f$'s lower semi-continuity will be enough. Then, we want to show that $f^{\text{A}}(\boldsymbol{x}_{\text{B}}; \boldsymbol{\theta}^{\text{A}})$ has lower semi-continuity when $(\boldsymbol{x}_{\text{B}}, \boldsymbol{y}_{\text{B}}) \in \mathcal{D}^{\text{B}}$. By the nature of adaptation regarding $\Delta(\boldsymbol{\theta}^{\text{A}} \backslash \boldsymbol{\theta}^{\text{B}})$,

$$f^{\text{B}}(\boldsymbol{x}; \boldsymbol{\theta}^{\text{B}^*}) = f^{\text{A}}(\boldsymbol{x}_{\text{B}}, \boldsymbol{\theta}^{\text{B}^*}) - f^{\text{B}\backslash\text{A}}(\boldsymbol{x}_{\text{B}}, \boldsymbol{\theta}^{\text{B}^*} \backslash \boldsymbol{\theta}^{\text{A}}), \tag{21}$$

when $f^{\text{B}\backslash\text{A}}$ is some function that satisfies the nature of adaptation.

Then, by Assumption 2 and the fact about the summation of lower semi-continuous functions, $f^{\text{A}}(\boldsymbol{x}_{\text{B}}, \boldsymbol{\theta}^{\text{B}^*})$ is continuous. Then, by the definition of $g$ and nature of composition of continuous functions, $f^{\text{A}}(\boldsymbol{x}_{\text{B}}, g(\boldsymbol{\theta}^{\text{B}^*})) = f^{\text{A}}(\boldsymbol{x}_{\text{B}}, \boldsymbol{\theta}_1^{\text{A}})$ also holds lower semi-continuity. Now, by Theorem 2 (Sec. A.2), $\hat{\boldsymbol{\zeta}} \to \boldsymbol{\zeta}^*$ almost surely. By Assumption 2, we get $||\boldsymbol{\zeta}|| < 1$, then almost surely, $\hat{\boldsymbol{\theta}}^{\text{bat}} \to \boldsymbol{\theta}^{\text{A}^*} = \varphi(\boldsymbol{\zeta}^*)$. Then by Theorem 3 (Sec. A.3) with Assumption 3, Assumption 4, and the argument above, the proof is completed. $\square$

## A.5 PROOF OF PROPOSITION 2

By Definition 2 (Sec. 3.3), one can derive from the assumption,

$$\frac{1}{k}||(\boldsymbol{H}^{\text{bat}|\text{A}})^{-1} \sum_{\mathcal{D}^{\text{bat}}} \nabla_{\boldsymbol{\theta}} \mathcal{L}^{\text{bat}|\text{A}}|| \leq \frac{1}{n}||(\boldsymbol{H}^{\text{bat}|\text{A}} - \boldsymbol{H}^{\text{bat}})^{-1} \sum_{\mathcal{D}^{\text{A}}} \nabla_{\boldsymbol{\theta}} \mathcal{L}^{\text{bat}|\text{A}}|| + o_P(1), \tag{22}$$

then, using the fact that $\mathcal{L}^{\text{bat}|\text{A}} \to \mathcal{L}^{\text{A}^*}$ by Proposition 1 (Sec. 1) and the nature of adaptation regarding $(\boldsymbol{\theta}^{\text{A}} \backslash \boldsymbol{\theta}^{\text{B}})$, one can derive that $\boldsymbol{H}^{\text{bat}|\text{A}} - \boldsymbol{H}^{\text{bat}} = \boldsymbol{H}^{\text{A}}$. With these facts,

$$\frac{1}{k}||(\boldsymbol{H}^{\text{bat}|\text{A}})^{-1} \sum_{\mathcal{D}^{\text{bat}}} \nabla_{\boldsymbol{\theta}} \mathcal{L}^{\text{bat}|\text{A}}|| \leq \frac{1}{n}||(\boldsymbol{H}^{\text{A}})^{-1} \sum_{\mathcal{D}^{\text{A}}} \nabla_{\boldsymbol{\theta}} \mathcal{L}^{\text{A}}|| + o_P(1). \tag{23}$$

is given. Then, by a using Newton's method, we can define,

$$\hat{\boldsymbol{\theta}}_n^{\text{bat}} - \boldsymbol{\theta}^{\text{A}*} = \frac{1}{k}(\boldsymbol{H}^{\text{bat}|\text{A}})^{-1} \sum_{K} \nabla_{\boldsymbol{\theta}} \mathcal{L}^{\text{bat}|\text{A}}, \tag{24}$$

$$\hat{\boldsymbol{\theta}}_n^{\text{A}} - \boldsymbol{\theta}^{\text{A}*} = \frac{1}{n}(\boldsymbol{H}^{\text{A}})^{-1} \sum_{G} \nabla_{\boldsymbol{\theta}} \mathcal{L}^{\text{A}}, \tag{25}$$

and with this, we can show that $\rho$ is

$$\mathbb{E} \operatorname{Tr}(\nabla_{\boldsymbol{\theta}} \mathcal{L} \nabla_{\boldsymbol{\theta}} \mathcal{L}^T \boldsymbol{H}^{-1} \boldsymbol{S} \boldsymbol{H}^{-1}), \tag{26}$$

and by combining the facts above, the theorem is proven. Also, recall that $\gamma \to 1$ will cause $\boldsymbol{H}^{\text{bat}} \to \boldsymbol{0}$ and $\sum_{\mathcal{D}^{\text{B}'}} \nabla_{\boldsymbol{\theta}} \mathcal{L}^{\text{bat}|\text{A}} \to 0$ by definitions proving the last part of the argument. $\square$

## A.6 PROPOSITION 1 FOR SPECIFIC ADAPTATIONS

**Proposition 1 for DreamBooth.** First, the loss function of DreamBooth is as follows:

$$\mathbb{E}_{x,c,\epsilon,\epsilon',t}\left[w_t\|\hat{x}_\theta(\alpha_t x + \sigma_t \epsilon, c) - x\|_2^2 + \lambda w_t'\|\hat{x}_\theta(\alpha_t' x_{\text{pr}} + \sigma_t' \epsilon', c_{\text{pr}}) - x_{\text{pr}}\|_2^2\right]. \tag{27}$$

$x$ is the latent that is going through the diffusion steps and $c$ is the text guidance. $\epsilon$ shows the noise prediction added in the latent each steps, $t$. Other variables are hyper-parameters to control the training (Ruiz et al., 2023a).

We can easily see that DreamBooth satisfies Assumptions 2, 3, and 4 of Proposition 1 (Sec. 4.1) as DreamBooth and diffusion model are considered to be learnable models. Let $\boldsymbol{\theta}^{\text{db}}$ and $\boldsymbol{\theta}^{\text{D}}$ represent

the parameters of DreamBooth and diffusion model correspondingly. Then, we observe that $\boldsymbol{\theta}_n^{\mathrm{db}}$ is a nearly minimizing estimator. Also, we see that

$$g(\boldsymbol{\theta}^{\mathrm{D}}) = \boldsymbol{\theta}_1^{\mathrm{db}} \Rightarrow g = \mathbf{1}_{\mathrm{identity}}, \tag{28}$$

as DreamBooth does not alter diffusion model parameters in the initializing step. Also, note that

$$g_2(\boldsymbol{\theta}^{\mathrm{D}}) = g(\boldsymbol{\theta}^{\mathrm{D}}) - \frac{\partial \mathbb{E}}{\partial \boldsymbol{\theta}^{\mathrm{db}}}, \tag{29}$$

for $\mathbb{E}$ is equation 27 which is shown to be continuous and by definition of partial derivation $g_2$ is continuous. We can use the same argument with all $g_n$ with $n > 2$. Thus, we have shown that $g$ is continuous, and by Proposition 1, DreamBooth can converge faster with backbone augmentation.

**Proposition 1 for LoRA.** Similar to the case of DreamBooth showing LoRA continuity will be sufficient to justify Backbone Augmented Training (BAT). To prove that LoRA is continuous, we need to show that the function $g(\boldsymbol{A}, \boldsymbol{B}) = \boldsymbol{W}_0 + \boldsymbol{A}\boldsymbol{B}$ is continuous. A function $g : \mathbb{R}^{d \times r} \times \mathbb{R}^{r \times k}$ and $\mathbb{R}^{d \times k}$ is continuous at $(\boldsymbol{A}_0, \boldsymbol{B}_0)$ if for every $\varepsilon > 0$, there exists a $\delta > 0$ such that:

$$\|(\boldsymbol{A}, \boldsymbol{B}) - \boldsymbol{A}_0, \boldsymbol{B}_0)\| < \delta \quad \text{implies} \quad \|f(\boldsymbol{A}, \boldsymbol{B}) - f(\boldsymbol{A}_0, \boldsymbol{B}_0)\| < \varepsilon.$$

The function $g(\boldsymbol{A}, \boldsymbol{B}) = \boldsymbol{W}_0 + \boldsymbol{A}\boldsymbol{B}$ involves matrix multiplication, which is continuous. The addition of $\boldsymbol{W}_0$ is constant and does not affect continuity. Hence, we need to show that the mapping $(\boldsymbol{A}, \boldsymbol{B}) \mapsto \boldsymbol{A}\boldsymbol{B}$ is continuous. Given small perturbations $\Delta\boldsymbol{A}$ and $\Delta\boldsymbol{B}$, we have:

$$g(\boldsymbol{A} + \Delta\boldsymbol{A}, \boldsymbol{B} + \Delta\boldsymbol{B}) = \boldsymbol{W}_0 + (\boldsymbol{A} + \Delta\boldsymbol{A})(\boldsymbol{B} + \Delta\boldsymbol{B}).$$

We expand the expression:

$$\boldsymbol{W}_{\mathrm{LoRA}} + \Delta\boldsymbol{W}_{\mathrm{LoRA}} = \boldsymbol{W}_0 + \boldsymbol{A}\boldsymbol{B} + \boldsymbol{A}\Delta\boldsymbol{B} + \Delta\boldsymbol{A}\boldsymbol{B} + \Delta\boldsymbol{A}\Delta\boldsymbol{B}.$$

The term $\boldsymbol{A}\Delta\boldsymbol{B} + \Delta\boldsymbol{A}\boldsymbol{B} + \Delta\boldsymbol{A}\Delta\boldsymbol{B}$ represents the change in $\boldsymbol{W}_{\mathrm{LoRA}}$ due to small perturbations in $\boldsymbol{A}$ and $\boldsymbol{B}$.

The perturbation $\Delta\boldsymbol{W}_{\mathrm{LoRA}} = \boldsymbol{A}\Delta\boldsymbol{B} + \Delta\boldsymbol{A}\boldsymbol{B} + \Delta\boldsymbol{A}\Delta\boldsymbol{B}$ can be bounded as:

$$\|\Delta\boldsymbol{W}_{\mathrm{LoRA}}\| \leq \|\boldsymbol{A}\|\|\Delta\boldsymbol{B}\| + \|\Delta\boldsymbol{A}\|\|\boldsymbol{B}\| + \|\Delta\boldsymbol{A}\|\|\Delta\boldsymbol{B}\|.$$

As $\|\Delta\boldsymbol{A}\| \to 0$ and $\|\Delta\boldsymbol{B}\| \to 0$, the perturbation $\|\Delta\boldsymbol{W}_{\mathrm{LoRA}}\| \to 0$. Therefore, for any $\epsilon > 0$, we can find a $\delta > 0$ such that if $\|\Delta\boldsymbol{A}\| < \delta$ and $\|\Delta\boldsymbol{B}\| < \delta$, then $\|\Delta\boldsymbol{W}_{\mathrm{LoRA}}\| < \epsilon$.

# B  EXPERIMENTAL DETAILS

In this section, we provide detailed explanations of the experimental setups and methodologies used in our study. Our experiments involve both diffusion model and language model to validate the propositions and evaluate the performance of various algorithms.

For the diffusion model (DreamBooth and LyCORIS), we used the LAION dataset (Schuhmann et al., 2022) as the backbone dataset $\mathcal{D}^{\mathrm{B}}$, since Stable Diffusion (Rombach et al., 2022) is pre-trained on it. We gathered adaptation datasets $\mathcal{D}^{\mathrm{A}}$ from sources like Textual Inversion (Gal et al., 2022) and Kaggle's 'Star Wars' dataset (Me, 2024). For the language model, we employed LLaMA 2-7B-alpaca-cleaned as the backbone language model. This model is LLaMA 2-7B (Touvron et al., 2023) specifically fine-tuned on the Alpaca-cleaned dataset (Taori et al., 2023b). Since most language models do not disclose their pre-training datasets, we adopted this publicly available model that had undergone further fine-tuning.

**DreamBooth.** For DreamBooth, all training was performed using a single NVIDIA RTX4090 GPU per adaptation. The typical learning rate was 5e-6. We used the AdamW optimizer for the entire training, with $\beta_1 = 0.9$ and $\beta_2 = 0.999$, a weight decay of 1e-2 andepsilon set to 1e-8. All inference seeds began with 42 and increased by 1 for each loop.

We gathered adaptation datasets from Textual Inversion (Gal et al., 2022), consisting of 5 images (e.g., red teapot and elephant datasets). DreamBooth's own dog dataset was also composed of 5 images. To construct the experiments, we generated optimal models with 40,000 to 50,000 denoising steps per dataset. BAT datasets were created by adding LAION data to the original datasets, and BAT training was conducted with these datasets.

**LyCORIS.** The LoCon algorithm, part of the LyCORIS library, introduces a low-rank adaptation technique specifically designed for convolutional layers in diffusion models like Stable Diffusion. Our experiments were conducted based on Stable Diffusion 1.4 as the backbone diffusion model (Rombach et al., 2022). Originally developed by (Hu et al., 2021) for attention layers in large language models, this adaptation for convolutional layers enhances image quality and fidelity during fine-tuning. For parameter-efficient fine-tuning (PEFT), we utilized LoCon among the LyCORIS methods. The learning rate was set to $5 \times 10^{-6}$, and the optimizer used was AdamW with $\beta_1 = 0.9$ and $\beta_2 = 0.999$. All training steps were fixed at 200, and a subset of these steps was plotted.

The dataset consists of movie character images sourced from a public dataset available on Kaggle, specifically the 'Star Wars' dataset (Me, 2024). Among the datasets used during the experiments applying LyCORIS PEFT, we focused on the characters Admiral Piett, Bodhi Rook, and Rose Tico. To train the optimal model and the BAT algorithm, we used different numbers of images per character. The optimal models for Admiral Piett and Bodhi Rook were trained on 91 images each, and Rose Tico's optimal model utilized 94 images. In contrast, the BAT algorithm used fewer images—10 for Admiral Piett, 43 for Bodhi Rook, and 38 for Rose Tico. When obtaining benchmark scores, we retrained the models with 300 training steps, keeping other experimental settings the same, and saved the model every 50 steps to extract the scores.

**LoRA & DoRA.** For LLaMA 2 based adaptations, NVIDIA A6000 GPUs are used according to the required experiments. LoRA's rank was set to 8. LoRA alpha was 32, and dropout was given by 0.1. Target model was query and value matrices of each transformer layer. The learning rate was 5e-5, and normally the batch size was 64. Weight decay was set to 0.01. We took MedQuad (Ben Abacha & Demner-Fushman, 2019), WinoGrande (Sakaguchi et al., 2021), and XSum (Narayan et al., 2018) as adaptation datasets $\mathcal{D}^\mathrm{A}$. To build the BAT set $\mathcal{D}^\mathrm{bat}$, we sampled $\mathcal{D}^\mathrm{B}$ at regular intervals and inserted the samples into $\mathcal{D}^\mathrm{A}$, also at regular intervals. Here, we set $|\mathcal{D}^\mathrm{A}| = 10000$ as a default.

## C    DATA SELECTION ALGORITHM

This is a general algorithm for data selection with $\mathcal{D}^\mathrm{bat}$ in our experiments. We considered those Hessian calculations as scores for each data referred in Kolossov et al. (2023). Rejecting data can be deemed as setting score to 0 like the data selection scheme covered in Sec. 2.

---

**Algorithm 1** Training Procedure for $\boldsymbol{\theta}^{\mathrm{A}^*}$ and $\boldsymbol{\theta}^{\mathrm{bat|A}}$

**Input:**
  $n \leftarrow |\mathcal{D}^\mathrm{A}|$ for the adaptation dataset; $k \leftarrow |\mathcal{D}^\mathrm{bat}|$ for the backbone augmented set
  $\mathrm{Score}_\mathcal{D}^\mathrm{A} := ||(\boldsymbol{H}^{\mathrm{bat|A}} - \boldsymbol{H}^\mathrm{bat})^{-1} \sum_{\mathcal{D}^\mathrm{A}} \nabla_{\boldsymbol{\theta}} \mathcal{L}^{\mathrm{bat|A}}||$ ; $\mathrm{Score}_\mathcal{D}^\mathrm{bat} := ||(\boldsymbol{H}^{\mathrm{bat|A}})^{-1} \sum_{\mathcal{D}^\mathrm{bat}} \nabla_{\boldsymbol{\theta}} \mathcal{L}^{\mathrm{bat|A}}||$

---

$i \leftarrow 1$
Train $\boldsymbol{\theta}^{\mathrm{A}^*}$
**while** Condition of Proposition 2 holds **do**
    Train $\boldsymbol{\theta}_i^{\mathrm{bat|A}}$
    $i \leftarrow i + 1$
    **if** $i \,\% n == 0$ **then**
        Calculate $\mathrm{Score}_\mathcal{D}^\mathrm{A}$
    **end if**
    **if** $i \,\% k == 0$ **then**
        Calculate $\mathrm{Score}_\mathcal{D}^\mathrm{bat}$
        **if** $\mathrm{Score}_\mathcal{D}^\mathrm{bat} \leq \mathrm{Score}_\mathcal{D}^\mathrm{A}$ **then**
            Continue
        **else**
            Select $\mathcal{D}^\mathrm{bat}$ again
            Go back to line 3
        **end if**
    **end if**
**end while**

---

## D  ADDITIONAL EXPERIMENTS

### D.1  METRICS

Using DINOv2 (Oquab et al., 2024), cosine similarity is used to measure the similarity between two feature vectors, often extracted from image representations. Given two vectors $\mathbf{v}_1$ and $\mathbf{v}_2$, their cosine similarity is computed as:

$$\text{Cosine Similarity}(\mathbf{v}_1, \mathbf{v}_2) = \frac{\mathbf{v}_1 \cdot \mathbf{v}_2}{\|\mathbf{v}_1\|\|\mathbf{v}_2\|}.$$

The centroid represents the mean vector of a set of feature vectors. The squared centroid is the square of the distance between the centroid and each data point. Suppose we have $N$ data points $\mathbf{v}_i \in \mathbb{R}^d$. The centroid $\mathbf{c}$ is given by:

$$\mathbf{c} = \frac{1}{N} \sum_{i=1}^{N} \mathbf{v}_i.$$

The squared centroid distance for each point $\mathbf{v}_i$ is:

$$\text{Squared Centroid Distance} = \sum_{i=1}^{N} \|\mathbf{v}_i - \mathbf{c}\|^2.$$

Where $\|\mathbf{v}_i - \mathbf{c}\|^2$ is the squared Euclidean distance between each point and the centroid. Lower centroid score shows that the output is more consistent with lower variance which infers better generalization.

CLIP uses cosine similarity to compare text and image embeddings. The model learns to maximize the similarity between matching text-image pairs while minimizing the similarity between non-matching pairs. Let $\mathbf{t}$ be the text embedding and $\mathbf{i}$ be the image embedding. The similarity score between them is calculated as:

$$\text{CLIP Similarity}(\mathbf{t}, \mathbf{i}) = \frac{\mathbf{t} \cdot \mathbf{i}}{\|\mathbf{t}\|\|\mathbf{i}\|}.$$

As $\mathbf{t} \cdot \mathbf{i}$ is the dot product between the text and image embedding, and $\|\mathbf{t}\|$ and $\|\mathbf{i}\|$ are the norms of the text and image embeddings. The cosine similarity is maximized for relevant text-image pairs and minimized for irrelevant pairs.

The Vendi score is a metric used to quantify similarity across multiple domains or datasets. It measures the overlap between sets of embeddings from different modalities (e.g., vision, text). Mathematically, Vendi score uses the concept of overlapping support across distributions.

Given two distributions of feature vectors $P$ and $Q$, the Vendi score can be formulated as:

$$\text{Vendi Score}(P, Q) = \int \min(P(x), Q(x)) dx.$$

This score evaluates how much of the support of one distribution is shared by the other, effectively measuring their similarity. Higher Vendi scores indicate greater overlap between distributions. Therefore, in the case of adaptations, lower Vendi scores implies the concentration of identity.

### D.2  RATIO TEST

In this section, we report the outcomes as we vary the proportion of the backbone data added in the adapter data $\mathcal{D}^A$. We selected $\gamma$ from 0.16 to 0.862 for DreamBooth adaptations trained with the same dataset and max iteration. All other settings are identical to those described in Sec. 5.1.2.

**Results.** The results of the ratio test are shown in Fig. 6. Notice that Proposition 2 mentions the convergence regarding not only training steps but also $\gamma$, the ratio of backbone and adaptation data. The proposition continues to imply that the convergence rate of $\gamma \to 0$ must be greater than the convergence of summation of loss gradient and Hessian matrix which represents the divergence of weights due to added backbone data. The experiments support this notion and exactly show that the increase of $\gamma$ is reducing the convergence rate of backbone augmented training.

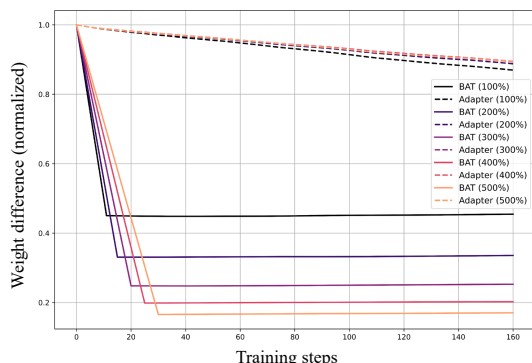

Figure 6: **Ablation on Backbone Augmentation Ratio.** The figure shows that DreamBooth adaptation's convergence rate is proportional to backbone augmentation ratio.

### D.3    OVERFITTING REGULARIZATION TEST

This experiment uses the same settings from Sec. 5.1.1.

**Results.** Proposition 1 shows the convergence of backbone augmented coefficient (Definition 2 in Sec. 3.3) which means that the case where backbone augmented training surpassing regular training is possible. This experiment intentionally induces overfitting as well to see whether the scheme regulates overfitting. Accordingly, we observe that convergence rate of the scheme is greater than regular training throughout total steps. Fig. 7 represents the outcome.

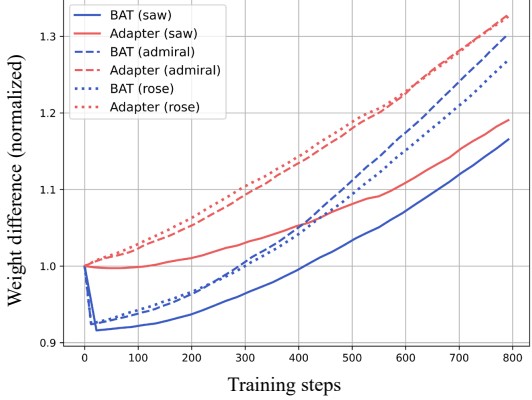

Figure 7: **Graph on Overfitting Regulation between BAT and Adaptations.** This figure shows the result of the overfitting experiment with full training steps. In various datasets, one can observe that BAT regulates overfitting better than regular DreamBooth adaptations.

### D.4 CHANGES IN STOCHASTIC BEHAVIOR

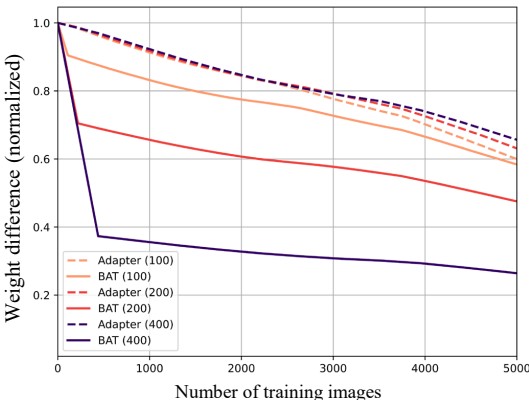

Figure 8: **Ablation Test regarding the Batch Size of BAT.** This test shows stochastic features are important for our method. One can see that that the convergence rate is proportional to the batch size. As the variety of input data is directly related to the performance of adaptations, we conjecture the batch size is related to the variety including the augmented backbone data.

### D.5 BAT WITH VARIOUS STARTING PARAMETERS

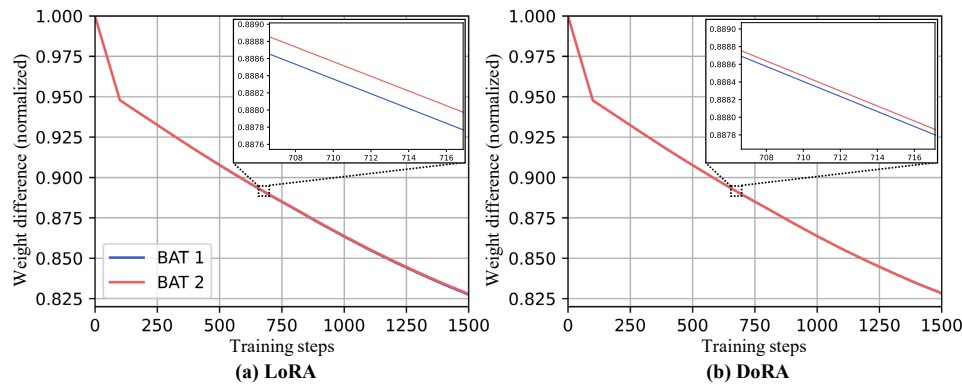

Figure 9: **Robustness in Deterministic Behaviors in Other Adaptations** This figure depicts the difference of convergence rate between our schemes with varying seeds. As language models have more parameters, the effect of non-deterministic feature reduces more comparing to diffusion adaptations.

## D.6    MORE QUALITATIVE ADAPTER RESULTS

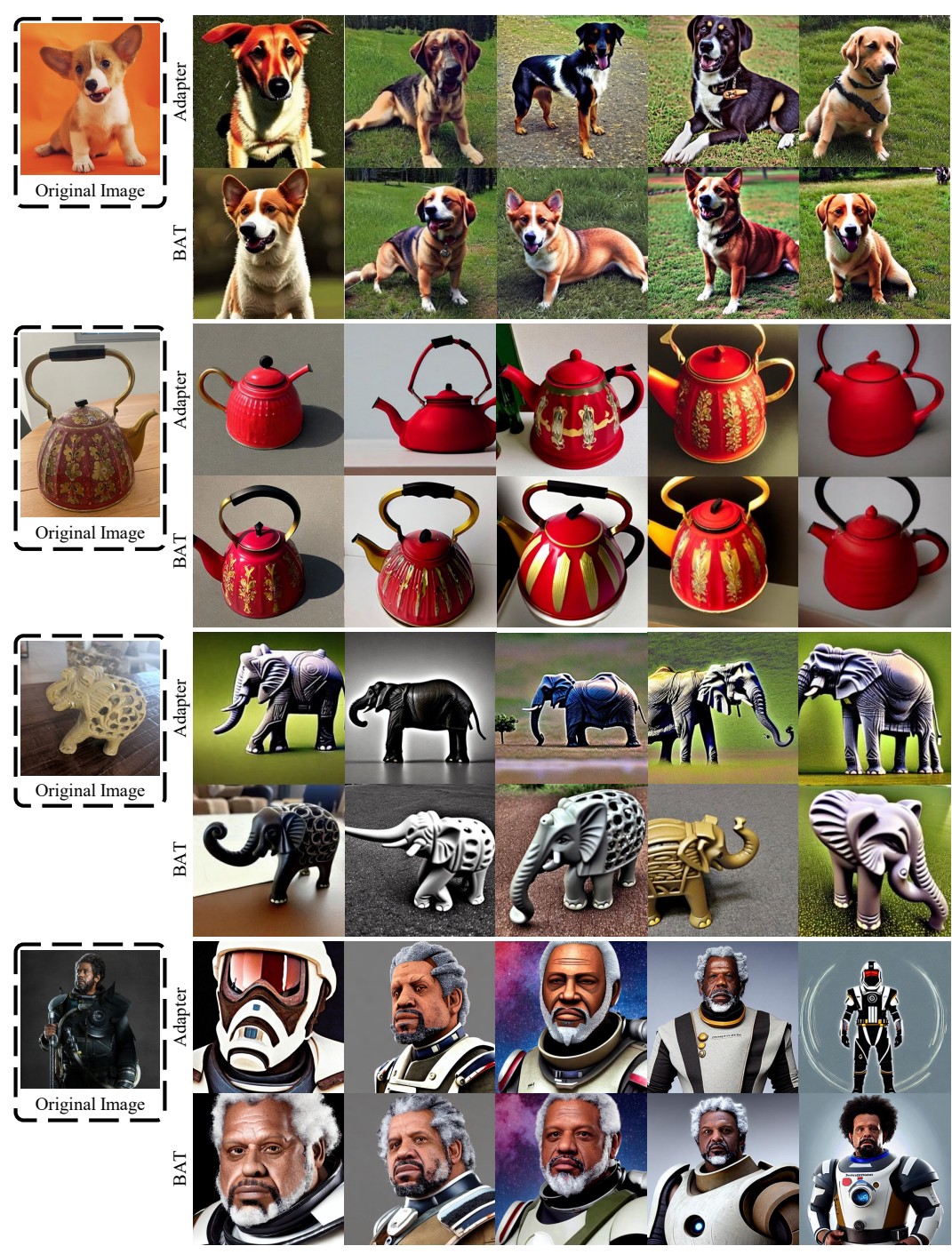

Figure 10: **DreamBooth Qualitative Outcomes.** These outcomes are gathered in the middle of DreamBooth training of a regular one and BAT. The purpose of this figure is to show the faster convergence rate of BAT over regular ones. Every class used the same models and every photo is simply a output of each model with a different random seed.

