# OpenReview forum: "BAT: Backbone Augmented Training for Adaptations"
_ICLR.cc/2025/Conference — ICLR 2025 Conference Withdrawn Submission_

### Official Review · Reviewer_y9wN · 2024-10-27

**Soundness:** 3
**Presentation:** 3
**Contribution:** 3
**Rating:** 6
**Confidence:** 3

**Summary:**

Context and Problem: The paper addresses the challenges faced during the adaptation of large generative models like GPT-3 and diffusion models, where adaptation techniques struggle with issues like mode collapse, knowledge shift, and high computational demands. Existing adaptation strategies either update the backbone model's parameters partially or modify them slightly, often leading to suboptimal performance and inefficiencies.

Solution Proposed: The core contribution is the Backbone Augmented Training (BAT), which integrates additional backbone training data into the adaptation process. BAT is designed to enhance the efficiency and effectiveness of adaptations by improving convergence rates towards optimal adaptation parameters. The paper provides theoretical support for BAT, demonstrating through mathematical proofs that BAT can achieve faster convergence than traditional methods. Empirical validation is also presented, comparing BAT to existing adaptation methods using metrics like cosine similarity and centroid distance on benchmark datasets.

**Strengths:**

Innovative Approach: BAT introduces a creative solution to the long-standing problem of inefficiency in model adaptations, providing a fresh perspective that leverages existing backbone data effectively.

Solid Theoretical Underpinning: The paper not only proposes a new method but also backs it with rigorous theoretical analysis.

Comprehensive Testing: Extensive empirical tests across different types of models (language and image) and datasets underline the method's versatility and robustness.

**Weaknesses:**

Potential Overfitting: There is a concern about the potential for overfitting, as BAT integrates more data from the backbone model, which might not always generalize well across diverse tasks.


BAT method require the existence of the backbone model data which might not be the case in some backbone models.

Quality of Backbone Data: The success of BAT heavily relies on the relevance and quality of the backbone data integrated during adaptation. Poor selection or low-quality backbone data could lead to ineffective learning or exacerbate existing issues like mode collapse and overfitting.

**Questions:**

How do you ensure that the backbone data integrated into BAT is of high quality and relevance to the specific adaptation task? Could you elaborate on any preprocessing or data selection criteria used?


In Figure 1, does a lower weight difference means better convergence rate?   In some tasks it would be better to deviate from the backbone data to solve the task at hand, relying on weight difference for the convergence rate feels like in continual learning where we try to limit weight updates to avoid catastrophic forgetting.

---

> ### Author Response · Authors · 2024-11-23
>
> We appreciate the reviewer’s thoughtful and constructive feedback on our paper. Below, we address the identified weaknesses and respond to your questions.
>
> ### Overall Explanation
>
> Overfitting is indeed a typical problem in adaptation training just as the reviewer points out. However, in our understanding, the diversity of train data tends to alleviate overfitting and this effect is more intense in adaptations settings. Some of the qualitative results show that the overfitting is regulated with BAT like in Fig. 7 in Line 1127. Therefore, overfitting will be mitigate with our method.
>
>
> ### About the lack of access to the backbone data
> - Many fine-tuned versions of these backbone models are available and are more likely to release their associated data. Utilizing this data facilitates the application of our propositions and the BAT framework.
> - Additionally, analyzing the input and output features of these models can help reveal the distributional similarities between accessible open-source data and closed-source data. This analysis could lead to modifications of our propositions, further demonstrating the feasibility of BAT. Developing mathematical proofs and conducting in-depth investigations in this direction would be a meaningful avenue for future research. To explore this possibility, we conducted experiments applying BAT using open-source data and a closed-source backbone model, with the results shown in Tab. 2.
>
> | Model                  | Cosine Sim (↑) | Centroid Distance (↓) | CLIP (↑) | Vendi (↓) |
> |------------------------|----------------|------------------------|----------|-----------|
> | DreamBooth              | **0.386**      | 797.78                | 0.267    | 4.812     |
> | + Similar BAT           | 0.365          | **795.78**            | **0.291**| **4.722** |
>
> ### Quality of Data
> The quality of backbone data is controlled by Proposition 2 and Algorithm 1 in Line 1006. This is one of our main contributions to prevent mode collapse and overfitting. We thank again for essential questions regarding our study.

---

### Official Review · Reviewer_KSSe · 2024-11-03

**Soundness:** 1
**Presentation:** 2
**Contribution:** 1
**Rating:** 5
**Confidence:** 2

**Summary:**

This paper tackles the problem of adapting foundation model training using backbone adapted training data. It introduces the BAT (backbone adapted training) method that selects training data for adaptation. It conducts some experiments with the weight difference metric to showcase results compared to a random selection baseline.

**Strengths:**

- The paper tackles an important problem surrounding adaptations in current foundation models, both unimodal and multimodal.

- The theoretical backing seems to be quite strong and well presented.

**Weaknesses:**

- Writing and Presentation: The overall presentation and story of the paper seem all over the place. It is not easy to follow any of the higher level points made in the theory section and how they connect with the empirical evidence. I would urge the authors to substantially rewrite parts of the results section to ease readability. For example, it is unclear how different parts presented in the theory section have been useful for the main empirical results presented.

- Lack of practically useful experiments / results: Most of the plots look at normalised weight differences, this isn’t particularly useful or exciting, are there results with the y-axis being some performance metric that is known to be a good measure of performance. For example, most adaptation methods test on tasks like MNLI, SST-2, MRPC, CoLA, QNLI etc for language and classification tasks like OxfordPets, ImageNet, DTD, Flowers etc for vision.

- Lack of appropriate baselines: All the results in the paper only compare adding BAT training to methods like LoRA and DoRA, however the improvements yielded by BAT are not compared to other standard adaptation baselines like VeRA, ETHER etc. Further, even the experiments conducted with LoRA and DoRA only measure the normalized weight differences which does not provide any signal on the effectiveness of the method in terms of downstream performance.

- Doesn’t the BAT method effectively use more unique samples for the adaptation data mixture? Wouldn’t that be a major confounder in the paper experiments — don’t we expect performance to get better as we include more relevant training data for the adaptation method? A more appropriate comparison would be to include a baseline that uses the same total number of training samples as BAT, but without selecting from the backbone data. This would help isolate the effect of the BAT selection method versus simply having more training data.

**Questions:**

- In some cases, the training data of the backbone is not known, for example GPT-4o, Llama-3.2, Gemini etc. In this case, we cannot explicitly do BAT since it relies on the training data of the backbone model right? Could the authors please discuss potential ways to approximate or estimate suitable backbone data when the original training data is not available? For example, are there ways to synthesize or curate proxy data that could still enable BAT-like approaches, while ensuring the overall data distribution remains preserved?

---

> ### Author Response · Authors · 2024-11-23
>
> We thank the reviewer for the insightful and constructive evaluation and questions on our paper. We replied on our weaknesses and your questions below.
>
> ### Writing and presentation.
>
> We regret the confusion with our presentation on this paper. By adopting the reviewer's opinions, we tried to modify the structure of the paper to make the reader to follow the arguments with ease. We agree that more connection between the propositions and the implications of them. We added explanations regarding them by dissolving former Sec.4.2.2. after we integrated the settings and definitions in Sec. 3. This will help the reader to navigate the study well as they may skip familiar parts. So, we updated the paper's notations to make them more descriptive. We separated the experiments into two main themes: theoretical validation and empirical results. Readers can refer to Sec. 5. We much appreciate the reviewer's advice.
>
> ### Lack of practically useful experiments / results.
>
> One of our paper’s main goals was to build a solid theoretical foundation for backbone augmented training as this method has a lot of room to develop in various applications. However, we agree that the implication of practicality of our theoretical method has to be more emphasized. Consequently, we replace Fig. 1 to demonstrate effectiveness of BAT compared to the former heuristic method. Also, we added experiment results from various metrics for language adaptations and visions in Fig. 1 and Tab. 1. We added common language adaptation metrics. Also, diffusion adaptation is added to the tables as well. Refer to Line 486 for more experiments.
>
> #### Table 2: Benchmark Comparison
> | Model  | BoolQ | PIQA  | SIQA  | HellaSwag | WinoGrande | ARC-e | ARC-c | OBQA  |
> |--------|-------|-------|-------|-----------|------------|-------|-------|-------|
> | LoRA   | 61.90 | 63.44 | 45.59 | 26.08     | 49.57      | 48.70 | **31.14** | **28.60** |
> | + BAT  | **62.42** | **74.48** | **70.21** | **27.48** | **51.38** | **53.20** | 30.55 | 25.40 |
>
> ### Lack of appropriate baselines.
>
> We thank the reviewer for suggesting new experimental approach for our study. With our limited resources, we tried to show that our method is applicable to the adaptations in both vision and language models in most comprehensive manner. Consider that our experiment is done also with vision adaptations such as DreamBooth and LyCORIS. VeRA and ETHER are indeed useful adaptations, however, we had to allocate our time and resource to demonstrate comprehensive manner of our method as we considered LoRA and DoRA are more prominent, then mentioned adaptations. Applying and modifying our method more specialize to other language adaptations seem to be an important future work. Yet, we aimed to show more details regarding the adaptation utilized in our studies which as shown below and integrated in Fig. 1. and Tab. 1.
>
> #### Table 1: Metrics Comparison
> | Model                  | Cosine Sim (↑) | Centroid Distance (↓) | CLIP (↑) | Vendi (↓) |
> |------------------------|----------------|------------------------|----------|-----------|
> | DreamBooth             | 0.386          | 797.78                | 0.267    | 4.812     |
> | + BAT                  | **0.418**      | **695.67**            | **0.315**| **2.191** |
> | LoCon                  | 0.5427         | 82.35                 | 0.4884   | 1.8471    |
> | + BAT                  | **0.5502**     | **82.48**             | **0.4952**| **1.8391**|
>
> ### Suggested experimental settings.
>
> The more appropriate comparison that the reviewer has suggested will increase the backbone augmented ratio, and just as in Sec. D.2. beginning in Line 1071, this will result only a better normalized weight. Yet, as we did not collected the metrics for this case. Just as the normalized weight indicated, we got the highest score with DreamBooth except text fidelity. We report this in the following:
>
> #### Table 1: Same amount of backbone and adaptation data.
> | Model                  | Cosine Sim (↑) | Centroid Distance (↓) | CLIP (↑) | Vendi (↓) |
> |------------------------|----------------|------------------------|----------|-----------|
> | DreamBooth             | 0.3811          | 667.7                | 0.273    | 2.787     |
> | + BAT                  | **0.4495**      | **632.3**            | **0.304**| **1.238** |
>
> This experiment validated our proposition once more. We thank the reviewer for this contribution.

---

> > ### Author Response · Authors · 2024-11-23
> >
> > ### Limited access to the backbone data.
> >
> > We recognize that many models used for adaptation do not make their data publicly available. To address this issue, we propose two potential solutions:
> >
> > - Numerous fine-tuned versions of these backbone models are available, and they are more likely to release their fine-tuned data. Leveraging such data enables the application of our propositions and the BAT framework.
> > - Also, one can analyze their input and output features. By examining these features, it is possible to demonstrate the distributional similarity between accessible open-source data and the closed-source data. This approach could allow for modifications to our propositions, demonstrating the feasibility of BAT. Developing mathematical proofs and conducting thorough investigations in this direction could be a valuable avenue for future research. To explore this possibility, we conducted experiments applying BAT with open-source data and a closed-source backbone model, with the results presented in Tab. 2.
> >
> > | Model                  | Cosine Sim (↑) | Centroid Distance (↓) | CLIP (↑) | Vendi (↓) |
> > |------------------------|----------------|------------------------|----------|-----------|
> > | DreamBooth              | **0.386**      | 797.78                | 0.267    | 4.812     |
> > | + Similar BAT           | 0.365          | **795.78**            | **0.291**| **4.722** |

---

> > > ### Comment · Reviewer_KSSe · 2024-11-25
> > > **Response to rebuttal**
> > >
> > > Having read through the author responses, some of my concerns have been alleviated but the majority of concerns still remain:
> > >
> > > (1) Thanks for including tab. 2 in the response, this validates my concern regarding downstream performance impact, however this is still only with language tasks. While a good starting point, I would recommend the authors to conduct a more systematic downstream performance analysis, also on vision tasks. This would significantly increase the quality of the paper in my opinion.
> > >
> > > (2) Thanks to the authors for conducting the additional experiments as suggested. However, my main concern is that these experiments are all again with the same distance measures rather than showcasing impact on downstream performance. In my opinion, all the ablations should also be conducted, both with the current metrics but also with downstream performance measures, since that is what practitioners care about at the end of the day. Alternatively, a substantial correlation analysis between the current distance measures and downstream performance might also suffice---however with the caveat that, such an analysis is only an indirect way to indicate performance on downstream tasks.

---

> ### Author Response · Authors · 2024-11-25
>
> We thank the reviewer again for the conducive opinion regarding our experiments.
>
> First of all, the reviewer recommends more experiments on vision tasks especially classification tasks. However, please remind that vision tasks like object classification and detection are known to have negative impacts when adaptations are applied. Large detection models such as DERT [1] and Swin [2] need utilize all the parameters and global context to achieve appropriate object detection and classification performance. Adaptations are not employed in these tasks typically. Please recall that our study is to improve the downstream tasks that recent adaptations target.
>
> In this context, we conducted more experiments concerning vision tasks than language tasks, especially in text-to-image generation and personalization which are the main stream of vision adaptations. Table 1: Metrics Comparison, Table 1: Same amount of backbone and adaptation data, and the table in Limited access to the backbone data are the results of those tasks. Vision adaptations other than generation related ones are relatively minor and independent part of the whole adaptation study.
>
> [1] End-to-End Object Detection with Transformers, Carion et al. 2020
>
> [2] Swin Transformer: Hierarchical Vision Transformer using Shifted Windows, Liu et al. 2021
>
> Finally, we wish the reviewer to clarify on that our additional experiments use **the same distance measure rather than showcasing impact on downstream tasks**. Please remind that Cosine Sim, CLIP, Vendi, and especially **Centroid Distance is not the weight difference between model parameters.** Centroid Distance calculates the distance between image pixels to validate the diversity of images. We generated 380 images per a vision adaptation in each step interval and calculated the expectation of those four metrics that are the most respected metrics in text-to-image generation so far. Please refer to [3], [4], [5] and [6] which are the studies regrading text-to-image generation adaptations that use the same metric to calculate their downstream performance.
>
> [3] DreamBooth: Fine Tuning Text-to-Image Diffusion Models for Subject-Driven Generation
>
> [4] An Image is Worth One Word: Personalizing Text-to-Image Generation using Textual Inversion
>
> [5] Navigating Text-To-Image Customization: From LyCORIS Fine-Tuning to Model Evaluation
>
> [6] The Chosen One: Consistent Characters in Text-to-Image Diffusion Models
>
> We sincerely hope that our response reduced the confusion on our former response and thank the reviewer again for thoughtful suggestion on our response.

---

### Official Review · Reviewer_XoLn · 2024-11-08

**Soundness:** 3
**Presentation:** 2
**Contribution:** 2
**Rating:** 3
**Confidence:** 3

**Summary:**

This paper explores the subject of training adapters for Diffusion models and Language models from limited data, and proposes a data mixing strategy called Backbone Augmented Training that mixes data from the set used in training the original model backbone, into the dataset used for learning adapters. The paper provided a theoretical justification for Backbone Augmented Training, proving that, for Dreambooth-based fine-tuning, and LoRA-based fine-tuning, the proposed method finds a set of weights that converges to an optimal solution on the adaptation tasks (if such a solution can exist).

The paper then explores the viability of the proposed method on learning tasks from the Dreambooth dataset, and a set of language tasks, including MedQuad, WinoGrande, and XSum. The experimental analysis focuses primarily on showing that the proposed method converges faster, according to a normalized weight difference comparing the training weights to those of an optimally trained model.

Experiments appear to show that BAT leads to consistently faster convergence across tasks.

**Strengths:**

Overall, results in the paper, while showing a promising initial signal for the method---the faster rate of convergence is indeed impressive---can be improved in a few key ways. For example, the selected set of datasets and tasks is a great start towards showing the viability of the method, but while measuring a normalized weight difference to a known good solution is helpful to see faster convergence, it is not by itself the most convincing method to showcase the quality of the found solution.

Recent works have found a surprising behavior for adapters, that the location in weight space of the adapter is unreliable at determining the quality of the adapter at a particular task:

[1] Prompt Waywardness: The Curious Case of Discretized Interpretation of Continuous Prompts, Khashabi et al. 2022.

[2] Understanding Visual Concepts Across Models, Trabucco et al. 2024.

For this reason, I am not convinced that a normalized weight difference is a sufficient evaluation metric to showcase the quality of the solutions found by Backbone Augmented Training, because there are very likely many near-optimal solutions dispersed throughout the weight space at different distances from the initialization of the adapter [2] that have comparable performance. To strengthen the evaluation, the authors could also report domain evaluation metrics, such as FID, and CLIP Scores for the diffusion-based tasks, and relevant NLP metrics for the language tasks.

---

## Originality:

The proposed method is to leverage samples from the pre-training dataset of the foundation model used to initialize the adapter, and mix this data with the target dataset for adaptation. The method is relatively simple as a result, but simplicity alone is not a weakness, and should be considered a strength in cases where the proposed method results in significant improvements in convergence speed and quality.

Adaptation of foundation models is becoming a well-studied problem, with LoRA being the de-facto in most cases, including the diffusion-based and language tasks explored in this paper. The problem statement explored in this paper is not particularly original, but the idea of leveraging the model’s original pre training data rather than generating samples from the model is original.

One important limitation of such an approach (and its theoretical analysis), is the reliance on accessing the model’s pre training data, which is becoming less true as many recent large-scale models are trained on closed-source datasets, including: Flux, SDXL, SD3+, Llama3+. How does the theoretical analysis extend to these cases, where researchers may not know what data the model observed in training?

---

## Quality:

Datasets and baseline in the paper are well selected.

The analysis appears to be of high quality, and the bridge from theory to empirical results by measuring the actual weight difference is a great start to showcase the faster convergence suggested by theory. However, the empirical results can be improved in a few key ways. First, reporting standard evaluation metrics, and showcasing faster convergence in these metrics during optimization is crucial to support the claim being made in this paper that BAT results faster convergence rates.

In addition to understanding the rate of convergence, an important experiment to understand the quality of the solutions found by BAT is to explore the FID vs CLIP Score pareto curve, and the FID vs Inception Score pareto curve, which showcase the tradeoff of image quality vs prompt adherence, and the tradeoff of image quality vs image diversity respectively. The strongest version of this paper would show that BAT systematically improves these pareto curves given fixed adaptation data, and compute budget.

As a final comment on quality, the results of base Dreambooth in Figure 2 are not convincing. The figure shows that Dreambooth-based adaptation is unable to learn to generate the corgi (left) which differs from results previously reported for Dreambooth on the corgi from [3], can this difference be explained?

[3] DreamBooth: Fine Tuning Text-to-Image Diffusion Models for Subject-Driven Generation

---

## Clarity:

The paper is generally organized well, and clearly communicates its ideas in most cases, excluding a few minor locations, where better phrasing would improve clarity.

Locations where clarity could be improved:

Line 152 (Equation 1) - variables are used without first being introduced.
Line 149-150 - shift and inject are used to describe optimizing the adapter weights for Dreambooth and Textual inversion methods, and I did not catch this on my first pass reading this section.

Line 206 - “parameters that are updated during adaptation” my understanding of this is that this set subtraction is the set of newly added parameters (and doesn’t include parameters from the original model updated as optimization progresses). This is an important distinction, because Dreambooth in its original formulation does not introduce new parameters per se, the entire model is fine-tuned, so this set is empty.

Line 223 - I’m not sure what is meant by `\hat{\theta}^{A}_{n + 1} = …` as my understanding is that the sets of parameters given by \hat{\theta}^{A} and \hat{\theta}^{A} \ \hat{\theta}^{B} are different.

Line 250 - I’m not sure what point is being made in this section that adaptation trainers “have to create the data in most cases” does this refer to Dreambooth-style prior preservation using synthetic examples?

Authors, please follow-up and clarify these sections.

---

## Significance:

Adaptation of foundation models is perhaps one of the most important tasks in modern deep learning, and faster optimization methods that require less data are an important research task. The problem statement is of great importance, and the goal of this paper to analyze and improve convergence rates is well timed. However, limited evaluation and lack of standard metrics, and results showing faster convergence of target metrics, rather than convergence of weight difference, limits the overall significance of the results.

**Weaknesses:**

Weaknesses have been woven into the strengths section, see above.

**Questions:**

Questions have been woven into the strengths section, see above.

---

> ### Author Response · Authors · 2024-11-23
>
> We appreciate the reviewer’s through and detailed investigation on our study. We continued our response concerning the suggestions and questions as follows.
>
> ### Usage of normalized weight difference.
>
> Just like you mentioned, global convexity and minimum cannot be determined easily in current deep learning models also in the adaptations in our studies. However, our assumptions on proposition, particularly Assumption 5 in Sec. 4.1, Line 280, presume local convexity and minimum which condone the adaptation training in the first place. Also, the optimal adaptation, which is the standard for normalized weight difference, not only went through the whole data, but also adopted optimal setting to achieve the best generation quality. So, the magnitude of proximity to the optimal weight(normalized weight difference) should justify as the settings for propositions at least. Nevertheless, we acknowledge that although this evaluation may satisfy the propositions, more practical tests are required to prove the advantages of our study. So, we calculated Cosine Similarity, CLIP Similarity, CDM, and Vendi Score for diffusion models, as we included LyCORIS from your suggestion, and more language model benchmarks. Please refer to Tab. 1 in Line 473. We also replaced Fig. 1 using practical metrics to compare former heuristic method and ours in Line 65.
>
> #### Table 1: Metrics Comparison
> | Model                  | Cosine Sim (↑) | Centroid Distance (↓) | CLIP (↑) | Vendi (↓) |
> |------------------------|----------------|------------------------|----------|-----------|
> | DreamBooth             | 0.386          | 797.78                | 0.267    | 4.812     |
> | + BAT                  | **0.418**      | **695.67**            | **0.315**| **2.191** |
> | LoCon                  | 0.5427         | 82.35                 | 0.4884   | 1.8471    |
> | + BAT                  | **0.5502**     | **82.48**             | **0.4952**| **1.8391**|
>
> ---
>
> #### Table 2: Benchmark Comparison
> | Model  | BoolQ | PIQA  | SIQA  | HellaSwag | WinoGrande | ARC-e | ARC-c | OBQA  |
> |--------|-------|-------|-------|-----------|------------|-------|-------|-------|
> | LoRA   | 61.90 | 63.44 | 45.59 | 26.08     | 49.57      | 48.70 | **31.14** | **28.60** |
> | + BAT  | **62.42** | **74.48** | **70.21** | **27.48** | **51.38** | **53.20** | 30.55 | 25.40 |
>
> ### Limited access to the backbone data.
>
> We understand that many models used for adaptation do not release their data. So, we suggest two solutions for this.
>
> - There are copious fine-tuned version of those backbone models which tend to release their fine-tuned data more often. Utilizing these data will allow the application of the propositions and BAT.
> - For closed-sourced datasets and large-scale models, one can examine their input and output features. By utilizing those examination, one may show the distributional similarity between accessible open-source data and closed-sourced data. This will lead to the modification of our propositions which proves the feasibility of BAT. Mathematical proof and through investigation in this approach might be a meaningful future work. To address this possibility, we conducted experiments that applied BAT with open-sourced data and closed-source backbone model as the result is shown in Sec. 5.2.2 in Line 486. We thank the reviewer again for the constructive suggestion. We add table for some of the result of this experiment.
>
> | Model                  | Cosine Sim (↑) | Centroid Distance (↓) | CLIP (↑) | Vendi (↓) |
> |------------------------|----------------|------------------------|----------|-----------|
> | DreamBooth              | **0.386**      | 797.78                | 0.267    | 4.812     |
> | + Similar BAT           | 0.365          | **795.78**            | **0.291**| **4.722** |

---

> ### Author Response · Authors · 2024-11-23
>
> ### Clarification on experiment settings.
>
> In the quality section, there is a comment about Fig. 2, the qualitative result of DreamBooth. We deleted this figure due to redundancy. You can find similar results in Section D.2 in Line 1188. Still, the question remains why the results display lower generation quality compare to former studies. Please note that our setting on this experiment is to show the faster convergence of BAT rather than the possibility of convergence. Just as you mentioned, DreamBooth showed the possibility of personalization for the first time, but in this experiment, we wanted to show that DreamBooth with BAT converges to the optimal adaptation faster than original DreamBooth taking equal amount of training steps. So, we displayed the results before the final convergence in the figure. Also, we applied the condition where we use insufficient amount of data for adaptation, which is our main problem throughout the study and this also limited the generation quality in both BAT and DreamBooth overall.
>
> Also, more investigation on practical metrics was demanded by the reviewer. Although we found this suggestion is helpful, FID is not often used in current text-to-image generation tasks, so instead, we utilized suitable metrics to depict suggested types of experiments. Please refer to [1] about this matter. Cosine Similarity represents the quality of the outputs. And CLIP score shows the fidelity to the text instruction. Vendi score and Centroid Distance are used to evaluation the personalization performance. Within the experiments, we observed there is no trade-off between the quality of outputs and text fidelity, and also there was no trade-off between the quality of images and personalization performance nor text fidelity and personalization. Please refer to Fig. 1 once more.
>
> [1] A Study on the Evaluation of Generative Models, Eyal Betzalel et al. 2022
>
> ### Clarification on notations and explanations.
>
> Regarding the loss function formula of DreamBooth, we added specific explanation on the variables in Appendix Sec. A.5 in Line 906. About the expression, DreamBooth is a personalization adaptation which repeatedly generates objects with same identity. Often, this personalization training is referred as injection of identity. But, there is also common overfitting problem of losing the ability to generate various objects and concepts in personalization which is called knowledge shift.
>
> Reviewer's point is legitimate on that DreamBooth does not introduce new parameters. However, DreamBooth is considered as an adaptation since it fixes input prompts, known as rare-token identifiers, which is associated with certain fixed parameters. So, for DreamBooth, the set subtraction would be the diffusion model and text encoder parameters that are not fixed rather than an empty set.
>
> We understand the set subtraction notation is not quite straightforward. We made a mistake of not adding the set subtraction on other variables. The renewed version is noted ((θ̂^A \ θ^B)_n+1) = ((θ̂^A \ θ^B^)_n) + Δ(θ^A \ θ^B)_n. This means that, for each training step, n, the adaptation updates only part of the backbone parameters and added adaptation parameters. For DreamBooth, they would be the unfixed parameters, and for LoRA, they would be additional decomposed matrices. Explanation is added on the paper as well. Further explanation is added in Line 191 and after.
>
> About the expression "have to create the data in most cases," we wanted to remind that adaptations demonstrate their significance best at the industrial level. Many trainers, end-users, especially, has to draw, design, and collect their adaptation data. However, we deleted this expression as we thought the disadvantage of the lack of data is emphasize enough.

---

### Official Review · Reviewer_drjm · 2024-11-11

**Soundness:** 2
**Presentation:** 1
**Contribution:** 2
**Rating:** 3
**Confidence:** 3

**Summary:**

Being able to efficiently adapt a pretrained large model for a specific task is extremely useful. The authors propose "Backbone Augmented Training," which selects training data from the original pretraining dataset to add to the finetuning dataset. Authors theoretically motivate using "backbone" training data and show that it approaches the optimal adaptation weights more quickly than naively ignoring the pretraining dataset or randomly selecting pretraining data to use.

**Strengths:**

- Authors connect their theory to existing methods, like Dreambooth and LoRA.
- Authors attempt to show results on both image (diffusion model) and language (Llama 2-7B) adaptation settings.

**Weaknesses:**

- The paper is very difficult to understand. There is a lot of unnecessary mathematical notation introduced, which obfuscates what is actually happening. Variables should be more descriptive (e.g. $\mathcal D_\text{pretrain}, \mathcal{D}_\text{finetune}$ instead of $\mathcal C, \mathcal G$). The main contributions of this work, Propositions 1 and 2, should have their full statements and assumptions in the main text. Each definition and theorem should have their meaning and importance explained intuitively in words.
- Evaluations are not convincing.
  - Why is normalized weight difference a good metric to show in the plots instead of test loss?
  - How much tuning has been done for the method vs the baselines?
  - Even if the proposed method converges slightly faster than the baseline, can't we take additional steps with the baseline to compensate, especially since we don't have to spend compute on data selection?

**Questions:**

- Fig 1: Why does random augmented training start high? Where is the comparison of loss, which is what we actually care about instead of "normalized weight difference"? Figure caption should be more descriptive, especially as it comes far before any explanation of the method or what the metrics mean.
- Eq. 1: none of these variables have been defined. Why is this necessary?
- L163: "We do not utilize these schemes or scores in the selection of backbone regularization data, but we follow a similar method in the experiments." Can you explain the specific differences between your method and previously proposed methods?
- L202: "$\theta^A := g(\theta^B)$: what is the intuitive meaning of $g$?
- L217: "compositional approaches between two risks are not valid for some cases." What does this mean?
- L263: what does "epoch error of $\mathcal C$" mean?
- What is an intuitive explanation for Proposition 2?
- Fig. 3: I'm confused by the schematic here. If the optimal $\theta^{A*}$ and $\theta^{bat | A*}$ do not coincide, how is this objective valid? Furthermore, can't we take larger steps with the original adaptation objective since it's flatter, and make the same amount of progress on the loss?
- Why do we care about normalized weight difference, especially since these problems are nonconvex and there are many good solutions?
- How does the proposed method compare against adding a regularization term on the model output, to encourage it to stay close to its pretrained outputs?

---

> ### Author Response · Authors · 2024-11-23
>
> We wish to thank the reviewer for the clear and helpful feedback and questions. We summarized our responses to the weaknesses and questions below.
>
> ## Regarding the clarity of our study
>
> ### Notations
> We acknowledge that our endeavor on clearer explanation was insufficient when our paper possesses a lot of mathematical materials and layers of arguments. As the reviewer required, we replaced the notations with more straightforward ones . Please see Sec. 3.2 starting from Line 185. Also, we added the context concerning those notations. Please refer to Line 227 to 231 and Line 237 to 242 in Section 3.3 as the example of such changes. Thank the reviewer again for this feedback. We also organized Definition section to let the readers jump to the novel and crucial part of this study when they are familiar with the basics. As we set the background for the mathematical generalization on adaptations, the function $g$ is defined as well, but this notation seemed misleading as the reviewer stated. This function is a mathematical model of loading an initialized adaptation over the backbone model. As we modify the notations, we added more descriptive explanation about the meaning of $g$ like the reviewer asked. Also, we moved the assumptions about our propositions to Sec. 4.1. from Appendix and added direct explanations about both propositions and assumptions. It starts from Line 260. These assumptions may answer the question regarding the nonconvexity and multiple solutions. Without these assumptions, theoretical exploration on models can be meaningless and impossible as these are weak assumptions that are utilized in many learning theories including [1]. Please refer to Theorem 3 in Appendix that already covers multiple solution cases in our assumptions with a global minimum space $\Theta^*$.

---

> > ### Author Response · Authors · 2024-11-23
> >
> > ### Figures and Equations
> > Regarding the question about Fig. 1 that the random augmented training starting high, we apologize about the confusion. We accidentally included the initial weight difference in the plot which is calculated before any adaptation training and BAT training as it should not be included in the plot. We replaced it with a correct one. Please refer to Fig. 3 in Line 486. We also added more description about the figure and the explanation of weight difference is added in Line 377. Also, about the necessity and details about DreamBooth loss function (Eq. 1.), the equation about loss function was needed to show our application of Proposition 1 in mathematical sense along with LoRA's example. However, as the reviewer stated, we found that more description about the equation would be helpful. We organized those examples and added explanation about notations and proofs in Sec. A.6. in Line 914. We appreciate this feedback. Proposition 2's equation is emphasized with space as the direct implication of this proposition is the basic criterion for our BAT data selection. Detailed explanation starts from Line 315.

---

> > > ### Author Response · Authors · 2024-11-23
> > >
> > > ### Confusing Expressions
> > > Some explanation about the former data selection such as "We do not utilize these schemes or scores in the selection of backbone regularization data, but we follow a similar method in the experiments." was unclear. We apologize for the misleading statement. Previously proposed method [1] selects their data by applying scores that will nullify the effect of input data whenever the data has lower score. Data selections in this study follows a similar fashion, but the former one was done in a semi supervised setting and they calculated the score before the training. Adaptations' data selections are done on fly and we simply remove the data from our batch without any direct loss manipulation. This explanation is added in Line 132. Also, about the question concerning the expression "compositional approaches", this means that the loss function of backbone and adaptations are using independent datasets which leads to a mathematical contradiction if one tries to use a mixed dataset and form a compositional function between backbone loss and adaptation loss without any justification. We again inserted clearer explanation starting from Line 214. Finally, the "epoch error of $\mathcal{C}$" was mentioned to distinguish between the subsampled error and epoch error in the former studies' format regarding estimators. The expression shows that the whole data in $C$ is calculated when S is defined as such. When S is different, the asymptotic may have other meanings. Clearer statement starts from Line 240.
> > >
> > > [1] Towards A Statistical Theory of Data Selection under Weak Supervision, ICLR 2024

---

> > > > ### Author Response · Authors · 2024-11-23
> > > >
> > > > ## Questions about our evaluations
> > > >
> > > > ### Justification on Our Metric
> > > > We thank again for the question that why we did not use the test loss as the metric. This was also a question brought out among ourselves. Nevertheless, first of all, please understand that we adopted this metric to demonstrate the coherence between our propositions and the actual application of adaptations, not to evaluate their performance. To alleviate this misunderstanding, we separated Sec. 5 into two experiments: Theory Validation and Performance Evaluation. Still, we carefully suggest that, for generative models such as diffusion and language models, the decreases of the test losses do not always imply the generation quality. However, our suggested optimal adaptations evidently possess better generation qualities as we trained them with known optimal settings. Thus, calculating the normalized weight difference between newly trained adaptations and the optimal adaptations represents the increase of generation quality better than comparing the test losses between them. Please refers to [2] and [3] on this detail. Lastly, as we agree on that more materials on evaluation are essential, we added more performance evaluations that are directly used in adaptation studies. Results are in Fig. 1 and Tab. 1.
> > > >
> > > > [2] How to Evaluate Generative Adversarial Networks, Brownlee et al. 2019
> > > >
> > > > [3] Evaluation Metrics for Generative Models: An Empirical Study, MDPI, 2022
> > > >
> > > > ### Explanations on Our Method
> > > > Several questions regarding our method were asked by the reviewer. We thank you again for the detailed investigation. On the questions concerning the amount of training between BAT and baselines, BAT does not training more than any baseline and their weight difference and performance metrics are calculated with the same amount of training. If the question was about the total training amount of both BAT and baselines compare to the former studies. Most of the experiments retain similar amount of training unless it is noted. However, it is true that our method requires additional calculation. So, we understand that if BAT only converge faster, why bother doing it when one can train adaptations longer without additional calculation? Yet, adding backbone data does not only accelerate the training process, but guide them to a better optimal point with the assumptions of Proposition 2. This also includes less overfitting and more preservation of prior knowledge. So, taking additional steps with regular adaptation training will not have the same effect with our method. This explanation is also related to the question regarding the Fig. 3 (now Fig. 2.), as the reviewer questions higher learning rate which has a similar effect to additional training steps. Just like you mentioned, taking larger steps in original adaptation will reach to the optimal point faster when the steps do not overfit, yet in the same steps whether they are larger or not, when Proposition 2 is satisfied with BAT, BAT will always surpass the original adaptation training. Our whole evaluation tries to support this claim including Fig. 1 and Tab. 2 and all the figures related to weight differences. Further, there is a distinction between other adaptation regulation as BAT will choose the data that will regulate the weight, if the initial weights need to be regulated; however, when the weights must be moved towards the optimal weights, BAT will enforce it. So, BAT not only regulates but also facilitates adaptation training.

---

### Note · Authors · 2025-05-16

I have read and agree with the venue's withdrawal policy on behalf of myself and my co-authors.

---

### Meta-Review · Area_Chair_yUED · 2024-12-23

**Metareview:**

This paper looks at the problem of adaptation of large backbone models, taking a theoretical perspective leading to the idea of selecting data used to train the backbone into the finetuning datasets. Empirical results are shown , mainly with respect to the metric of normalized weight difference, across a number of tasks including vision and language. Reviewers appreciated the theoretical motivation of the paper and simplicity of the method, but had a number of reservations collectively and most reviewers expressed that these put the paper below the bar. For example, the clarity of the writing and theory was mentioned by several reviewers (important given that it is the main contribution), the use of normalized weight difference as the main metric for evaluation was questioned by several reviewers (and indeed some generative results were seen as poor), several baselines were missed, etc. While we appreciate the rebuttal provided by the authors, which addressed some of the concerns, as mentioned by reviewer KSSe many concerns remain.

  Considering all of the materials, I agree with the majority of reviewers and do not recommend acceptance. While the theory and practice of adaptation is hugely important, the paper requires significant modification both in terms of the theory (ideally providing more intuition and more understandable exposition) and practice (with more rigorous results and use of better/more standard metrics) are needed.

**Additional Comments On Reviewer Discussion:**

Many concerns were raised by the reviewers. Some, such as use of public data for closed backbones, were addressed but many were not as mentioned by reviewer KSSe.

---

### Decision · Program_Chairs · 2025-01-22

Reject